# Probing the functional impact of sub-retinal prosthesis

**Sébastien Roux[1]\*, Frédéric Matonti[1,2], Florent Dupont[3,4], Louis Hoffart[1,2], Sylvain Takerkart[1], Serge Picaud[5], Pascale Pham[3,4], Frédéric Chavane[1]\***

[1]Institut de Neurosciences de la Timone, CNRS, Aix-Marseille Université, Marseille, France; [2]Ophthalmology Department, Aix Marseille Université, Hôpital Nord,Hôpital de la Timone, Marseille, France; [3]CEA-LETI, Grenoble, France; [4]Université Grenoble Alpes, Grenoble, France; [5]Inserm, UMRS-986, Institut de la vision, Paris, France

**Abstract** Retinal prostheses are promising tools for recovering visual functions in blind patients but, unfortunately, with still poor gains in visual acuity. Improving their resolution is thus a key challenge that warrants understanding its origin through appropriate animal models. Here, we provide a systematic comparison between visual and prosthetic activations of the rat primary visual cortex (V1). We established a precise V1 mapping as a functional benchmark to demonstrate that sub-retinal implants activate V1 at the appropriate position, scalable to a wide range of visual luminance, but with an aspect-ratio and an extent much larger than expected. Such distorted activation profile can be accounted for by the existence of two sources of diffusion, passive diffusion and activation of ganglion cells' axons *en passant*. Reverse-engineered electrical pulses based on impedance spectroscopy is the only solution we tested that decreases the extent and aspect-ratio, providing a promising solution for clinical applications.

## Introduction

Blindness affects 45 million people around the world with an increase of 1 to 2 million people each year (*Resnikoff et al., 2004*). The two main retinal pathologies are age-related macular degeneration (AMD, *Ambati and Fowler, 2012*; *Finger et al., 2011*) and Retinitis Pigmentosa (RP, *Bocquet et al., 2013*). Although the genetic alterations and the mechanisms subtending photoreceptor death are well described, and therapeutic strategies are under clinical trials (*Ferrari et al., 2013*; *Talcott et al., 2011*), retinal degeneration inexorably leads to blindness (*Tsujikawa et al., 2008*). In this perspective, retinal prostheses provide a promising solution that remains to date a unique alternative for the patients. Restoring some visual perception using implants has been already achieved (*Humayun et al., 2012*; *Shepherd et al., 2013*; *Zrenner et al., 2010*) but still offers insufficient gains in visual acuity (*Humayun et al., 2012*; *Zrenner et al., 2010*); (*Nanduri et al., 2012*; *Rizzo, 2003*). Despite having fundamentally different designs and operating modes, the two main models of prostheses proposed to the patients with RP (Argus II epiretinal prosthesis and the subretinal alpha IMS microphotodiode array) restore some visual function although with a spatial resolution (*Ahuja and Behrend, 2013*; *Humayun et al., 2012*); (*Stingl et al., 2013*) that does not allow for the recognition of faces or autonomous locomotion. Improving the performances of such implants is thus a key strategic issue for further developments.

This problem is actually a general issue shared by other sensory prosthesis, such as the cochlear implants (*Kral et al., 1998*). This latter field, well in advance compared to retinal implants, has already demonstrated the importance of developing animal models for better understanding of the underlying physiological processes (*Fallon and Shepherd, 2009*; *Miller et al., 2000*). To improve the resolution and efficiency of retinal prosthesis, it is therefore necessary to launch appropriate

**\*For correspondence:** sebastien. roux@univ-amu.fr (SR); frederic. chavane@univ-amu.fr (FC)

**eLife digest** One of the most common causes of blindness is a disorder called retinitis pigmentosa. In a healthy eye, the surface at the back of the eye – called the retina – contains cells called photoreceptors that detect light and convert it into electrical signals for the brain to process. In people with retinitis pigmentosa, these photoreceptor cells die off gradually, which leads to loss of vision.

The only treatment available for retinitis pigmentosa is to have an artificial retina implanted into the eye. The artificial retina consists of an array of tiny electrodes, which take over from the damaged photoreceptors and generate electrical signals. The person with the implant perceives these electrical signals as bright flashes called "phosphenes". However, the phosphenes are too large and imprecise to provide the person with vision that is good enough for tasks such as walking unaided or reading.

To find out why artificial retinas produce such poor resolution, Roux et al. compared how a rat's brain responds to either natural visual stimuli or activation of implanted an array of micro-electrodes. Both the micro-electrodes and the natural stimuli activated the same areas of the brain. However, the micro-electrodes produced larger and more elongated patterns of activation. This is because the electrical currents generated by the micro-electrodes diffused throughout the retinal tissue and activated other neurons besides those intended. To overcome this problem, Roux et al. tested different ways of stimulating the micro-electrodes in order to identify those that induce the desired patterns of brain activity. This approach – known as reverse engineering – did indeed improve the performance of the micro-electrode array.

The next step is to extend these findings, which were obtained in healthy rats, to non-human primates or animal models of retinitis pigmentosa to better understand the condition in humans. In addition, combining the current approach with other existing techniques should further improve the vision that can be achieved with artificial retinas.

models that allow probing precisely and quantitatively the functional impact of prosthetic activation. Pioneering animal studies have proposed to use cortical recordings to explore the efficiency of various patterns of retinal electrical stimulation (*Chowdhury et al., 2008*), including current steering methods (*Jepson et al., 2014a*; *Matteucci et al., 2013*) or the effect of the return electrode configuration (*Cicione et al., 2012*; *Matteucci et al., 2013*; *Wong et al., 2009*) and studied the temporal aspect of prosthetic vision (*Elfar et al., 2009*; *Fransen et al., 2014*; *Jepson et al., 2014a*, *2013*, *2014b*; *Nadig, 1999*; *Schanze et al., 2003*; *Sekirnjak et al., 2008*; *Wilms et al., 2003*). However, most were not designed to characterize and calibrate the functional activation of the visual system (*Chowdhury et al., 2008*; *Eger et al., 2005*; *Mandel et al., 2013*; *Nadig, 1999*; *Schanze et al., 2003*; *Walter et al., 2005*; *Wong et al., 2009*) nor probed the functional impact of implants through systematic comparison with visual activation (*Chowdhury et al., 2008*; *Cicione et al., 2012*; *Eger et al., 2005*; *Fransen et al., 2014*; *Mandel et al., 2013*; *Matteucci et al., 2013*; *Schanze et al., 2003*; *Wong et al., 2009*). Hence, none of these studies allowed to fully address the question of understanding and controlling the functional impact of retinal prosthesis.

To address this issue, we developed an acute animal model to quantitatively assess the functional impact of retinal prostheses by comparing the downstream activation of the visual system in response to visual *versus* artificial stimuli, using intrinsic optical imaging of the primary visual cortex (V1). To infer the hypothetic visual counterparts induced by electrical stimulation, we first established a quantitative mapping of the cartographic organization of the rat visual system. So far only Gias and colleagues (*Gias et al., 2004*) have provided a retinotopic description of the rat visual cortex using optical imaging. Here, we generalized this mapping to visual parameters that are the most important for prosthetic vision (position, size and intensity). Using this cartographic benchmark, we demonstrate that prosthetic stimulation generates a functional activation that occurs at the expected retinotopic location and amplitude. However, the aspect ratio and the extent of the activation are significantly larger than expected. This can be explained through a simple model with two sources of diffusion: an electrical passive diffusion and the activation of axons en passant from ganglion cells.

To control the extent of cortical activation, we tested various patterns of electrical stimulation and showed that only reverse engineering of the electrical pulses to inject the desired electrical stimulation (*Dupont et al., 2013*; *Pham et al., 2013*) allowed focalizing the activation. This result provides a promising perspective that could be easily implemented for improving the visual acuity of already implanted patients.

## Results

In order to investigate the functional impact of acutely implanted retinal prostheses, we have recorded the population activation of V1 of anesthetized rats using optical imaging (*Eckhorn et al., 2006*; *Gias et al., 2004*; *Grinvald et al., 1999*; *Walter et al., 2005*) (*Figure 1A–B*) and implanted micro-electrode array prosthesis (MEA, see Materials and methods) sub-retinally (*Figure 1C*). With such preparation, we quantified for each rat the visually as well as the artificially-evoked cortical population activations. Our results thus are derived from a systematic pairwise comparison of the artificial activation versus its counterpart visual activation (*Figure 1A*) in response to relevant key parameters (position, size, shape and intensity) across a large number of animals (N = 35). Hence,

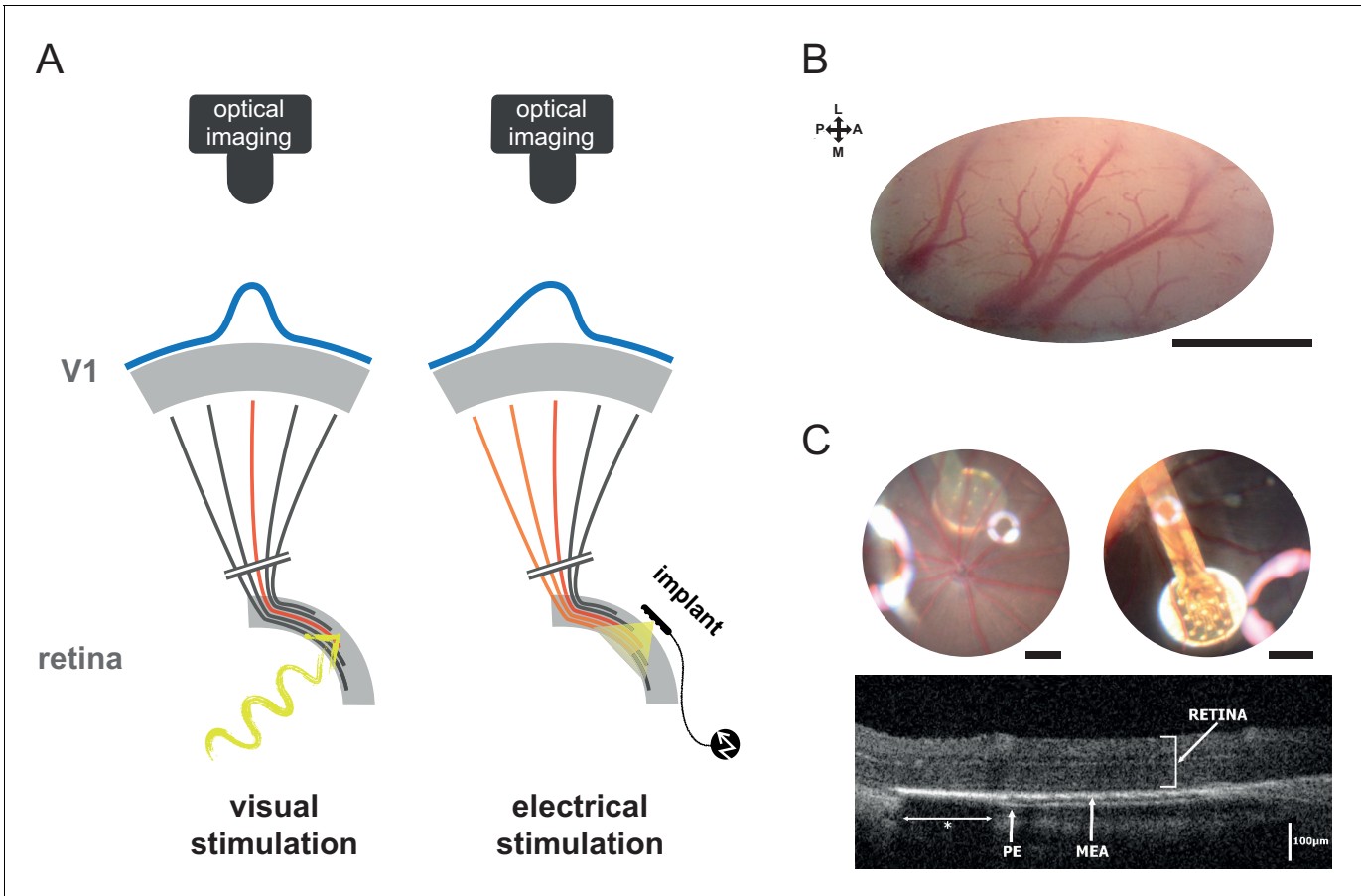

**Figure 1.** Experiment design. (**A**) Schematic view of the experimental setup with the camera and the visual pathway from the retina to V1 activated with normal visual stimuli (left) or with sub-retinal electrical stimulation using a MEA (right). Retinal ganglion cells' (RGCs) axons leaving the retina and projecting to V1 via the LGN are schematized (black when non activated; in red for direct activation; in orange for activation of en passant fibers that could occur in the electrical stimulation case). Blue curves on the top of V1 schematize the expected spatial profile of cortical activation (symmetric for visual stimulation and asymmetric for electrical stimulation if axons en passant are activated). (**B**) Clear optical access through thinned bone over V1; scale bar: 2 mm. L: lateral; M: medial; A: anterior; P: posterior. (**C**) Image of the eye fundus with the 9 electrodes (top left) and the 17 electrodes (top right) MEA; scale bars: 500 microns. Note that the use of an additional magnifying lens induced optical artifacts (halos of light). Retinal OCT B-scan (bottom) of an implanted animal showing the MEA and intact retina (PE: pigmetary epithelium; *: shadowing of the external reference surrounding the 17 electrodes MEA).

responses to visual stimuli were used to provide a functional benchmark to which prosthetic activations were systematically compared.

## Position

First, we investigated whether retinal prosthesis stimulation generates activation at the expected retinotopic position. To answer this question, we mapped the cortical retinotopic organization of the rat visual cortex (*Gias et al., 2004*) by flashing white squares in a 5x4 grid of 20° side (*Figure 2A*). For each animal, we computed polar maps for azimuth and elevation (*Figure 2B,F* for 2 different animals). Animals were also implanted sub-retinally with a MEA and its retinal position was identified using fundus imaging (*Figure 2C,G*), reported in the visual (A,E) and cortical domains (B,D,F,H), respectively. In the first animal (*Figure 2* top row), we stimulated the whole MEA (wMEA), the size of which (1 mm) was in the range of the size of the stimuli used for visual retinotopic mapping (20°)

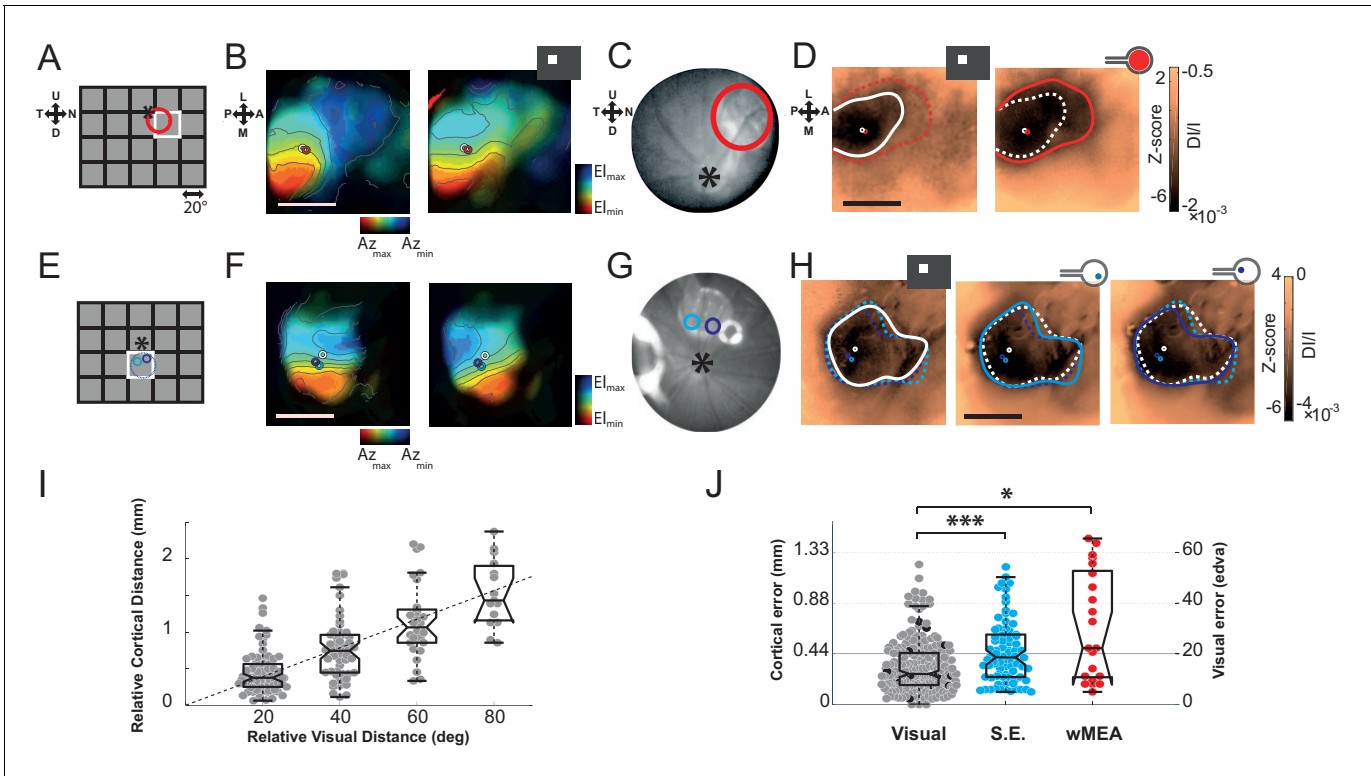

**Figure 2.** Position. The visual (**A&E**), retinal (**C&G**) and cortical (**B,D&F,H**) expected position and size of the MEA are compared to their corresponding visual stimuli. (**A**) Schematic view of the visual field showing the 20 positions (grid) of the visual stimuli used for retinotopic mapping. Optic disk: asterix; MEA: colored circles; nearest visual stimulus: white square (stands for all Figures). (**B**) V1 retinotopic polar map for azimuth (left) and elevation (right). Color hue and brightness code respectively for the retinotopic position and the strength of the response. Scale bar: 2 mm. (**C**) Image of the eye fundus with the implant. (**D**) Extent and center-of-mass (contour and circle respectively) of V1 activations generated by visual (white, right map) and stimulation at ± 150 µA wMEA (red, left map). The activation amplitude depicted in the colorbar is expressed both in Z-score and in DI/I (see Materials and methods). The solid line indicates the activation contour of the corresponding map and the superimposed dashed line corresponds to the compared condition; scale bar: 2 mm. (**E-H**) Same as in (**A-D**) in a different animal for 2 SE stimulations (cyan and purple; dashed circle: MEA position) at ± 200 µA; see *Figure 6A* for an example at low intensity in the same animal. (**I**) Retino-cortical magnification factor for azimuth computed over 20 animals (n = 177 displacements). Boxplots represent the median and interquartile range; whiskers represent ± 2.7σ or 99.3 coverage if data are normally distributed (any points outside are considered as outliers). (**J**) Retinotopic based positional cortical error for visual and electrical activations (right: mm; left: equivalent degrees of visual angle). Black dots correspond to visual counterparts of electrical activations. One-sided two-sample Wilcoxon rank sum test for paired data: $p_{SE\ vs.\ wMEA}$=0.056, $n_{wMEA}$ = 21, $n_{SE}$ = 82, $N_{wMEA}$ = 6, $N_{SE}$ = 7. Wilcoxon rank sum test for paired data: *p=0.025, n = 13, N = 4 and ***p=5.35 10⁻⁵, n = 80, N = 6 (*p<0.05, **p<0.01, ***p<0.001, n = number of sample, N = number of rats).

The following figure supplement is available for figure 2:

**Figure supplement 1.** Raw maps.

(*Hughes, 1979*; *Palagina et al., 2009*). *Figure 2D* right shows the cortical activation generated by such electrical stimulation at ± 150 µA (red contour). For a better visualization, we provided all the maps without contour in Supplementary *Figure 2—figure supplement 1* with scale bars expressed both in Z-score (left of the colorbar) and in DI/I (right of the colorbar). Note that Z-score and DI/I measures were highly and significantly correlated (median correlation coefficient $r^2 = 0.81$ between all pixels of Z-score *vs.* DI/I maps with the corresponding [20-50-80] percentiles being [0.73–0.81–0.88]%, all $p_{val} = 1.40e^{-45}$, N = 9 rats, n = 225 maps). A 20° visual stimulus presented at this position (*Figure 2A* white square) generated activation at a similar position (*Figure 2D* left, white circle) but with an extent (white contour) that was much smaller than its electrical counterpart (red dashed line). We then tested single electrode stimulation (SE, see Materials and methods). *Figure 2E–H* shows an example in another animal for two individual electrodes each stimulated at ± 200 µA. The evoked activations for the two SE (*Figure 2H* middle and right maps, cyan and purple) were about the size of the 20° visual activation (left map, white) and their positions within the activation generated by the corresponding visual stimulus covering their retinal positions (*Figure 2E*). To generalize these observations, we computed the retino-cortical magnification factors by measuring the cortical distance between the centers of activation (white circles) elicited by visual stimuli displayed at different positions. Across 20 animals, the cortical distance between activations was plotted against the distance separating the visual stimuli (*Figure 2I*). We observed a linear increase of cortical distance with visual distance, at a rate of 22 µm/° (in accordance with the literature) (*Gias et al., 2004*). For all activations (visual and electrical), we used this value to estimate the error (in mm and equivalent degree of visual angle, °eq) between the position of the activity's center-of-mass and its expected position within the retinotopic map. Visually evoked activations provided an approximation of the inherent variability of that measure (*Figure 2J*, gray dots). On average, the cortical error was 0.32 ± 0.22 mm for visual stimuli, 72% of the data points falling within the retinotopic representation of the 20° stimulus (<0.44 mm). The same measure was applied to electrically evoked activations for SE and wMEA stimulations. For SE, the estimation of the position of the evoked activity was quite accurate (error of 0.46 ± 0.27 mm), 55% of evoked responses falling within the retinotopic expected position. Note that these error values are actually over-estimated since in practice, the implant was not obligatory located at the exact same retinal position that corresponds to the visual stimulus. Finally, wMEA activations yielded responses with more variable and less accurate position than SE with an average error of 0.68 ± 0.47 mm, 38% of the evoked positions being within the expected representation.

## Size

In the previous examples, we observed that the electrically-evoked activations are larger than expected (*Figure 2D,H*). We thus quantified this effect by systematically comparing the extents of the visually and artificially evoked activations (*Figure 3*). As shown in the example of *Figure 3A*, the extent of cortical activation linearly increased with visual stimulus size with an average slope of 81 µm/° when estimated at population level (*Figure 3B*, N = 7). In comparison, the active surface diameters of the SE and wMEA were of 0.05 and 0.65 mm (see Materials and methods) equivalent to 1 and 11° of visual angle (*Hughes, 1979*), respectively. Although the visual stimuli covered a larger region of the retina (20°, *Figures 2D* and *3C* top right), we observed a systematic and highly significant increase of the cortical activation extent induced by wMEA stimulation in 94% of the cases (17/18 conditions, 10 animals; see Materials and methods and *Figure 3D* legend for details on statistical procedures), with an average increase of 2.9 ± 2.01 times the activation extent of the visual stimulation (*Figure 3D*, thin gray lines). Please note that for this analysis, all electrical stimulations were systematically paired with the closest visual stimulus used for retinotopic mapping (*Figure 3C*, bottom maps). In comparison, SE stimulation led to an average activation size of 0.9 ± 0.5 times their 20° visual counterparts (*Figure 3D*). Using the visual-size tuning function (*Figure 3B*), we extracted the equivalent visual stimulus sizes that would have evoked such cortical activations. These values reached on average 24.4 ± 10.6 and 32.2 ± 11.5° instead of 1 and 11° for the SE and wMEA configurations, respectively. Electrical stimulation hence led to extremely large diffusion of the activation, comparatively larger for SE than wMEA. Such differences could be explained by several factors including (i) the extent of wMEA activation being sometimes underestimated because of the limit of the imaged region of interest and (ii) the close proximity of the annular counter-electrode (reference) in wMEA constraining lateral diffusion of the current (*Cicione et al., 2012*; *Pham et al., 2013*; *Wong et al., 2009*). Our results unambiguously demonstrate that prosthetic retinal stimulation

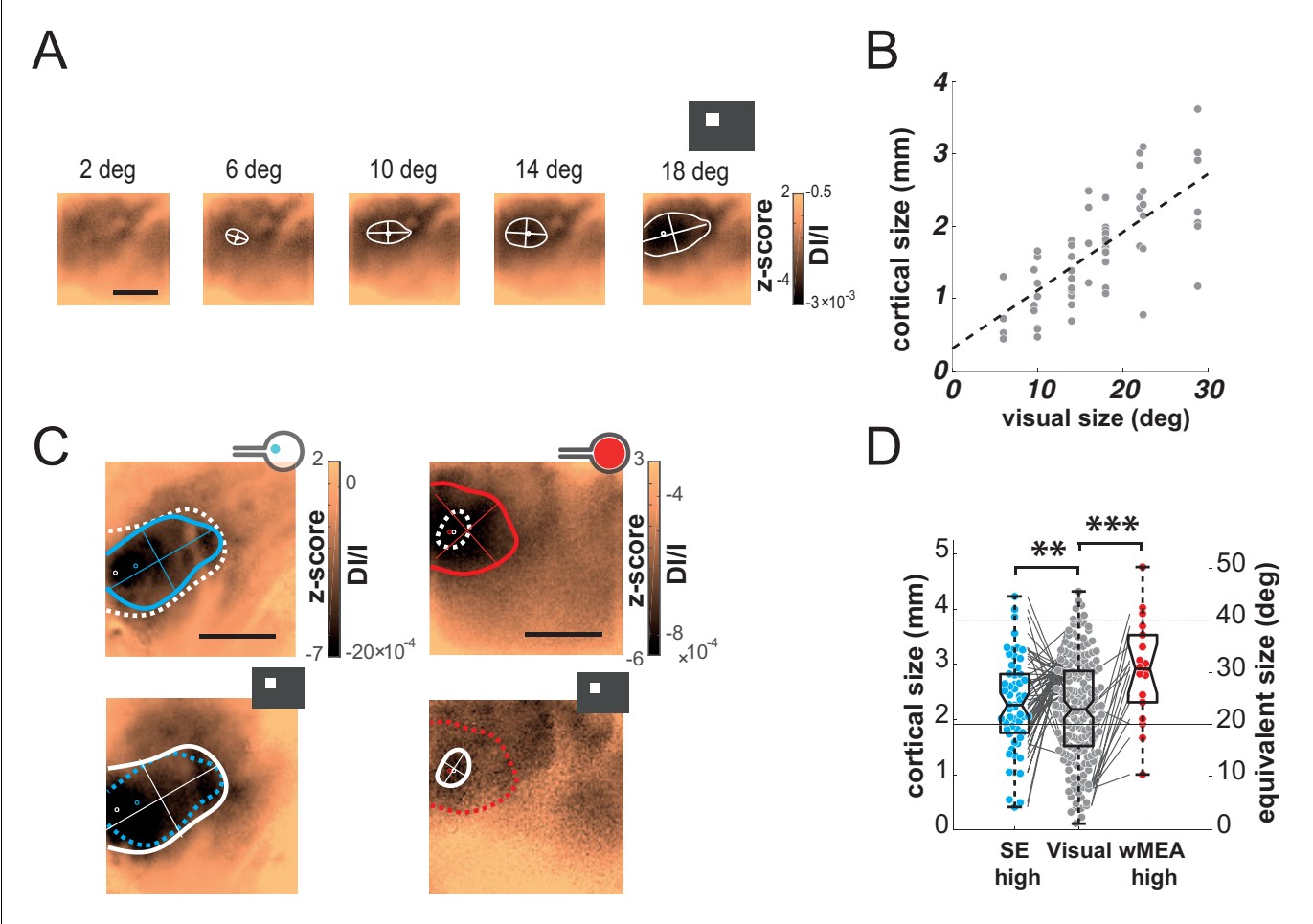

**Figure 3.** Size. (**A**) V1 activation generated by visual stimuli of increasing size (value indicated above the maps). Center-of-mass: white circle; extent: white contour; equivalent ellipse orientation: white cross; scale bar: 2 mm. (**B**) Extent of cortical activation as a function of visual stimulus size pooled over 7 rats (linear fit: dashed black line). (**C**) Extent of cortical activations generated in 2 animals by SE (blue) and wMEA (red) stimulation (top maps) at a high current intensity (± 200 and 150 µA respectively) and their corresponding 20° visual stimulus (white, bottom maps). Centers of mass of the activation: colored circles; scale bar: 2 mm. (**D**) Size of cortical activation generated by visual (gray, N = 20 rats); SE (blue, N = 8 rats) and wMEA (red, N = 10 rats) stimulation. Wilcoxon rank sum test for paired data: **p=0.0019 (n = 54, N = 8), ***p=1.83 $10^{-4}$ (n = 15, N = 10). An alternative ordinate of equivalent visual counterpart is given on the right. Gray thin lines link paired electrical stimulation to visual activation. Solid horizontal black line indicates the size of a 20° visual stimulus estimated from the fit in B. Note that we could not reveal any effect of the electrode-to-counter-electrode distance on activation size in the various SE configurations (not shown).

generates too large activation to obtain the necessary independence between electrodes; an inter-electrode spacing of at least 0.8 to 1.2 mm on the retina, at low- and high-intensity respectively, would be required to yield non-overlapping cortical activations.

## Shape

Next we compared the shape of cortical activations to visual and prosthetic stimulations, because it may help understanding the origin of the spread. When compared to visual stimulus (white), we generally observed elongated cortical activation for SE (*Figure 4A* blue) but not for wMEA (red) stimulation. In order to quantify this elongation, we computed the aspect ratio (AR) of activation contours at the level of the population (*Figure 4B*, see Materials and methods). Using pairwise visual-prosthetic comparison, we found a highly significant (see *Figure 4B* legend) increase in the aspect ratio of the evoked activations for SE condition compared to their visual counterparts (1.66 ± 0.43 *vs.* 1.35 ± 0.19) but none for wMEA. These elongations could be caused by the recruitment of ganglion cells'

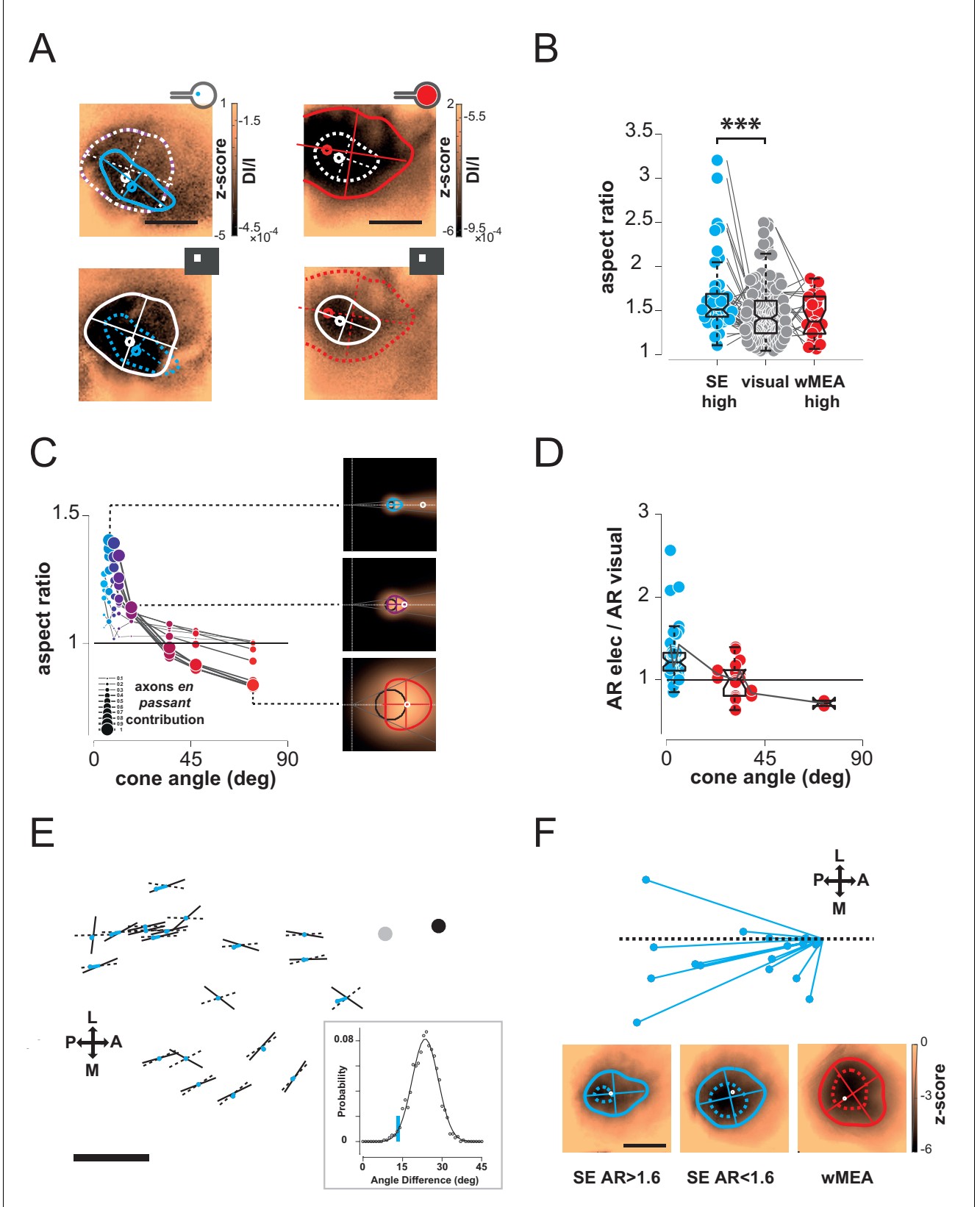

**Figure 4.** Shape. (A) Shape of cortical activations generated in 2 animals by SE (blue) and wMEA (red) stimulation at high current intensity (top) and their corresponding 20° visual stimulus (white, bottom). (B) Aspect ratio (AR) of cortical activations (Wilcoxon rank sum test for paired data, ***p=1.06

*Figure 4 continued on next page*

*Figure 4 continued*

$10^{-4}$, n = 44, N = 7). (C) Predictions of the elongation of electrical activations as a function of the contribution of axons en passant and the distance to the optic disk. Insets correspond to a model of retinal activation due to direct isotropic activation plus passive electrical diffusion and anisotropic activation due to axons en passant recruitment for 3 different electrode sizes. The brightness codes the strength of the response. Center of the white dashed target: position of the optic disk; black circle: position and size of the MEA active surface; gray lines: 'shadow cone' angle sustained by the MEA active surface respective to the optic disk location; colored contour: size and shape of the global retinal activation for an axons en passant contribution of 1 (alpha, see Materials and methods). (D) Elongation of electrical activations relative to their corresponding visual activations (AR electrical/AR visual) as a function of the 'shadow cone' angle. (E) Cortical radial organization of prosthetic activations. Solid segments: orientation of cortical activations; dashed segments: optimal radial orientation towards the black disk; segment crossing; geometrical center; red dot: center-of-mass of cortical activations; Dark disk: cortical position that optimized radial organization; gray disk: median position of the optic disk. The blue lines connect the center-of-mass to the geometrical center of activations. Scale bar: 0.5 mm. Inset: distribution of median angular deviation expected by chance compare to our observation: blue segment. (F) Top: centered and reoriented deviations of the center-of-mass (blue disks) to the geometrical center (center of the representation), horizontal dashed axis corresponds to the orientation of the radial organization. Bottom: averaged, centered and reoriented SE (with AR > and < than 1.6, left and middle respectively) and wMEA maps (right). White circle: center-of-mass; additional dashed contour corresponds to a Z-score of −4.5.

The following figure supplements are available for figure 4:

**Figure supplement 1.** Model of retinal anisotropic activation.

**Figure supplement 2.** Model of retino-cortical transformation.

axons *en passant* leading to oriented anisotropic cortical activations (*Figure 1A* right). However, why would SE activation yield stronger AR than wMEA? One explanation is that the ganglion cell neurons with axons that will be activated *en passant*, all converging radially towards the optic disk, are all located upstream to the electrode within a 'shadow cone' with a top angle that will (i) decrease with the distance between the implant and the optic disk location and (ii) increase with the size of the stimulated retinal surface (*Figure 4—figure supplement 1*). In a simple functional model (see Materials and methods), we combined the predictions of what should be the shape and size of retinal activations due to local and *en passant* activation. Isotropic 'direct' activations were modeled as a Gaussian activation around the site of stimulation (*Figure 4—figure supplement 1* left column) for different electrode sizes (*Figure 4—figure supplement 1*, rows). Anisotropic *en passant* recruitment of ganglion cell axons was modeled as a shadow cone activation, i.e. all peripheral ganglion somata whose axons have been activated by the electrical stimulation (*Figure 4—figure supplement 1*, middle column, *Figure 4—figure supplement 1A* for details in the model, see Materials and methods). *Via* a weighted sum, we combined these two activations (right column and *Figure 4C* insets) with different ratios (*Figure 4—figure supplement 1B*: 0.5 & C: 1). From these predicted activations, we extracted similar parameters (size, position, elongation) as we did from our cortical recordings. Since our interest is to predict the effect of axon-en-passant activation on radial elongation, we used the following convention for the AR: the numerator is the length of activation along the radial axis and the denominator is the length of activation along a tangential axis (perpendicular to the radial axis). This model predicted that the resulting retinal activation should indeed be more elongated along radial axis for smaller cone angle (*Figure 4C* and *Figure 4—figure supplement 1*, AR > 1), this effect being stronger when we increase the relative contribution of en passant activation. The model further predicts that the retinal activation should become more elongated along a tangential axis for very large cone angle (AR < 1). Please note that these predictions are left unchanged if we apply a retino-cortical transformation to the simulated activations (see *Figure 4—figure supplement 2C*). To test for these predictions, we plotted the aspect-ratio of the activation across animals and experimental conditions as a function of the retinal 'shadow cone' angle formed by the stimulated surface (*Figure 4D*). Please note that, to account for any potential deformation of the evoked activity due to retino-cortical magnification factor (see *Figure 4—figure supplement 2A*) or physiological noise, we normalized all electrically-induced AR to their corresponding visual AR. In this plot we observed a similar decrease of the aspect ratio with the cone angle, confirming our predictions (*Figure 4C*). Our results thus suggest that the difference we observed between wMEA and SE is explained if we make the hypothesis that part of the functional activation of the visual system comes from of axons *en passant* recruitment. Note that our model suggests that the

isotropy observed for wMEA activations simply results from the geometrical arrangement between the electrode size and the shadow cone. We thus conclude that wMEA activations are still suffering from contamination by the activation of fibers en passant. Hence SE simulation remains the best configuration to achieve the highest performance in spatial resolution. According to our simple model, the relative contribution of axons *en passant* to the global prosthetic activation should be of the same order of magnitude as direct activation (alpha = 1, see Materials and methods: model of retinal activation).

Thus, the predictions that arise from this observation are: for small cone angle (i) the elongation of the cortical activation should be radially organized towards the representation of the optic disk and (ii) the activation should be anisotropic, attracted by the radial elongation towards more eccentric positions (i.e. away from the optic disk representation); lastly (iii) for large cone angle, cortical activation should be more elongated along a tangential axis. To test for these predictions, we pooled together all activation profiles that were sufficiently elongated, i.e. with an AR above 1.6, and plotted them in cortical space (*Figure 4E*). In this figure, each solid segment corresponds to the orientation of a cortical activation. The dark disk is the cortical position that optimized the radial arrangement of the observed orientations (see Materials and methods). Observed orientations of cortical activation deviated from an optimal radial organization (dashed lines) to this cortical position by only 13.3° (median value across N = 20 activations). Importantly, this cortical position is close to, and in the same direction as, the median position of the optic disk (gray disk). This latter one was estimated from 16 experiments whenever its mapping was possible. However, we could not use it systematically because of imprecision in its estimation (see Materials and methods). We then checked whether this result could occur by chance. For that purpose, we generated 1000 random distributions of orientations at the observed positions and, for each, looked for the cortical position that optimized radial arrangement (as described above). For this position, we computed the median angular deviation between the random orientations and the optimal radial arrangement to this position. This led to a Gaussian distribution of median angular deviation expected by chance in our particular configuration, with a mean of 23.6 ± 4.9° (*Figure 4E* inset). Our observation of 13.3° (red line) is significantly smaller than what would be expected by chance (p=0.02, see Materials and methods). To test for the second prediction of anisotropic activation that our model raised, we compared the center-of-mass of the activation (*Figure 4E* red dot) to the geometric center (dashed and solid line crossing). Indeed if the activity is attracted away from the blind-spot, the center-of-mass must be delocalized from the geometrical center, opposite to the position of the blind-spot. We plotted the position of the center-of-mass (red points in *Figure 4E*), and indeed observed that it was pushed away from the optic disk representation. *Figure 4F* present a centered, reoriented and zoomed view on all deviations of the center-of-mass (red disk) to the geometrical center (center of the representation) within the same reference frame (horizontal dashed axis being the radial organization). Center-of-masses all deviated away from the optic disk with an averaged angular deviation of −168° (opposite deviation being 180°). To better illustrate this deviation, we averaged all the maps under consideration in this analysis, after realignment along the radial axis (the axis linking the activation to the BS, here the BS is to the right) and centering (*Figure 4F* bottom left). We can see from this averaged map that there is indeed a radially elongated and anisotropic activation opposite to the representation of the blind spot (dotted contour represent activation higher than 4.5 z-score). For comparison, we made similar averaging on all the other maps in response to SE (with an aspect-ratio < 1.6) and wMEA. For SE activation with low AR, the result shows an activation that is more isotropic. For MEA activation, the averaged map shows, as predicted, a slight elongation along the tangential axis. Hence, in our experimental observations, the radial arrangement and deviation of the center-of-mass of the activation behave as expected from an activation of en-passant axons in the retina.

## Intensity

Encoding different levels of luminance is an important aspect of prosthetic vision. We thus investigated the effect of stimulus intensity. Increasing luminance non-linearly increased the cortical activation size and amplitude (*Figure 5A*). Across the population (N = 8), we fitted the response amplitude (*Figure 5C*) with a Naka-Rushton function (*Naka and Rushton, 1966*) for the whole population (black dotted line), as well as for each animal individually (thin gray lines). The luminance of semi-saturation was 9.6 ± 2.9 cd/m$^2$. We then compared these results with the intensity of electrical stimulation (*Figure 5D–E*) that similarly modulated the cortical activations (*Figure 5B*). Over 35

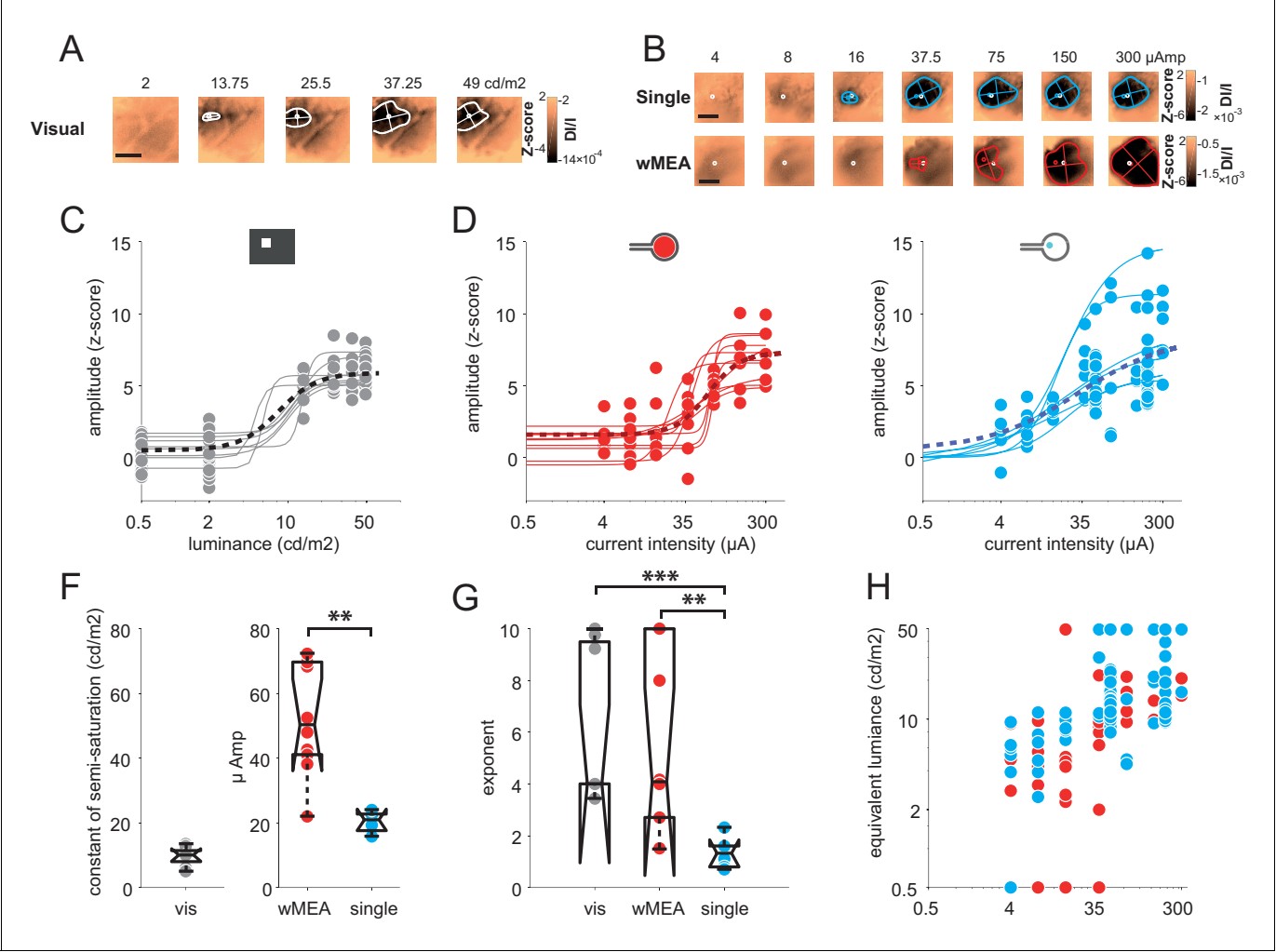

**Figure 5.** Intensity. (**A**) V1 activation generated by visual stimuli of increasing luminance (value indicated above maps). Center-of-mass: white circle; extent: white contour; equivalent ellipse orientation and size of activation contour: white cross; scale bar: 2 mm. (**B**) V1 activation generated by SE (top) and wMEA (bottom) in 2 different animals at high current intensity. Center-of-mass of visual and electrical activations are indicated with white and colored circles respectively and extent of electrical activation in colored contours; scale bar: 2 mm. (**C**) Amplitude of cortical activation as a function of visual stimulus luminance computed over 8 rats (population fit: dashed black line; individual fits: gray thin lines). (**D**) Amplitude of cortical activations generated by wMEA stimulation at high current intensity. Population fit: dark red dashed line; individual fits: orange thin lines (N = 10). (**E**) Amplitude of cortical activations generated by SE stimulation at high current intensity. Population fit: dark blue dashed line (N = 25). Note that individual fits (cyan thin lines) were only performed on N=6 animals (tested with 7 different levels of intensity) and not on the others (N=19, tested with only 2 different levels: 50 and 200 μA). (**F**) Constant of semi-saturation (c50) for visual (gray in cd/m$^2$) and electrical activations (wMEA: red and SE: blue, in μA). Two-sample Wilcoxon rank sum test: **p=0.0017, $n_{SE} = N_{SE} = 6$, $n_{wMEA} = N_{wMEA} = 10$. (**G**) Exponent of the naka-rushton fits for visual (gray) and electrical activations (wMEA: red and SE: blue). Two-sample Wilcoxon rank sum test: ***p$^{SE\ vs.\ visual}$=6.66 10$^{-4}$ ($n_{SE} = N_{SE} = 6$, $n_{visual} = N_{visual} = 8$); **p$^{wMEA\ vs.\ SE}$=0.0075 ($n_{SE} = N_{SE} = 6$, $n_{wMEA} = N_{wMEA} = 10$); p$^{wMEA\ vs.\ visual}$=0.896 ($n_{wMEA} = N_{wMEA} = 10$, $n_{visual} = N_{visual} = 8$). (**H**) Amplitude-based correspondence for all electrical activations to their visual counterpart (wMEA: red and SE: blue) as a function of current intensity level (in equivalent cd/m$^2$).

animals (wMEA: 10, SE: 25), we observed a gradual increase of response amplitude for both wMEA and SE electrical stimulation, well captured by Naka-Rushton fits (*Figure 5D–E*, thick dotted lines). The intensity of semi-saturation (*Figure 5F*) was significantly lower for SE than for wMEA (20.29 ± 3.25 μA *vs.* 52.66 ± 17.35 μA, see legend for details on statistical procedure). *Figure 5G* compares the exponent of the individual fits performed on the amplitude parameter for visual, SE and wMEA stimulation. Steep transitions were observed for wMEA (n = 5.58 ± 3.54) and for visual luminance (n = 6.05 ± 3.01) whereas more gradual transitions were observed for SE (n = 1.34 ± 0.6) which significantly differed from wMEA and visual stimuli (see *Figure 5G* legend). As a consequence, the

operating range (see Materials and methods) of the intensity response function in visual stimulation was on average 9.2 ± 4.8 cd/m², 82.5 ± 88.4 μA for wMEA and much broader for SE (182.6 ± 151.4 μA). This is further reflected in *Figure 5H* plotting the hypothetic equivalent luminance of electrical activations. We estimated the equivalent visual luminance that corresponds to the response amplitude evoked by electrical stimulation using the Naka-Rushton fit (*Figure 5C*). The functional operating range induced by current intensity manipulation allowed generating an artificial activation with an equivalent luminance that could theoretically be in the range of 2–50 cd/m².

## Focalization

The previous results show that artificial activations of the visual system with sub-retinal implants generate stimuli of appropriate position and scalable intensity, but whose size is on average 2.4 and 5.8 times larger than expected for wMEA and SE, respectively. This clearly impairs the functional efficiency of the implants. We therefore took advantage of our experimental design to seek solutions for controlling the spatial extent of the functional activation. Several candidate parameters have been investigated for SE stimulation (*Figures 6* and *7*, respectively blue vs. green colors): polarity (cathodic vs. anodic pulse first, *Figure 6A–B*), symmetry vs. asymmetry of the biphasic square pulse (*McIntyre and Grill, 2000*) (*Figure 6C–D*) and regular pulses vs. electrical impedance spectroscopy-

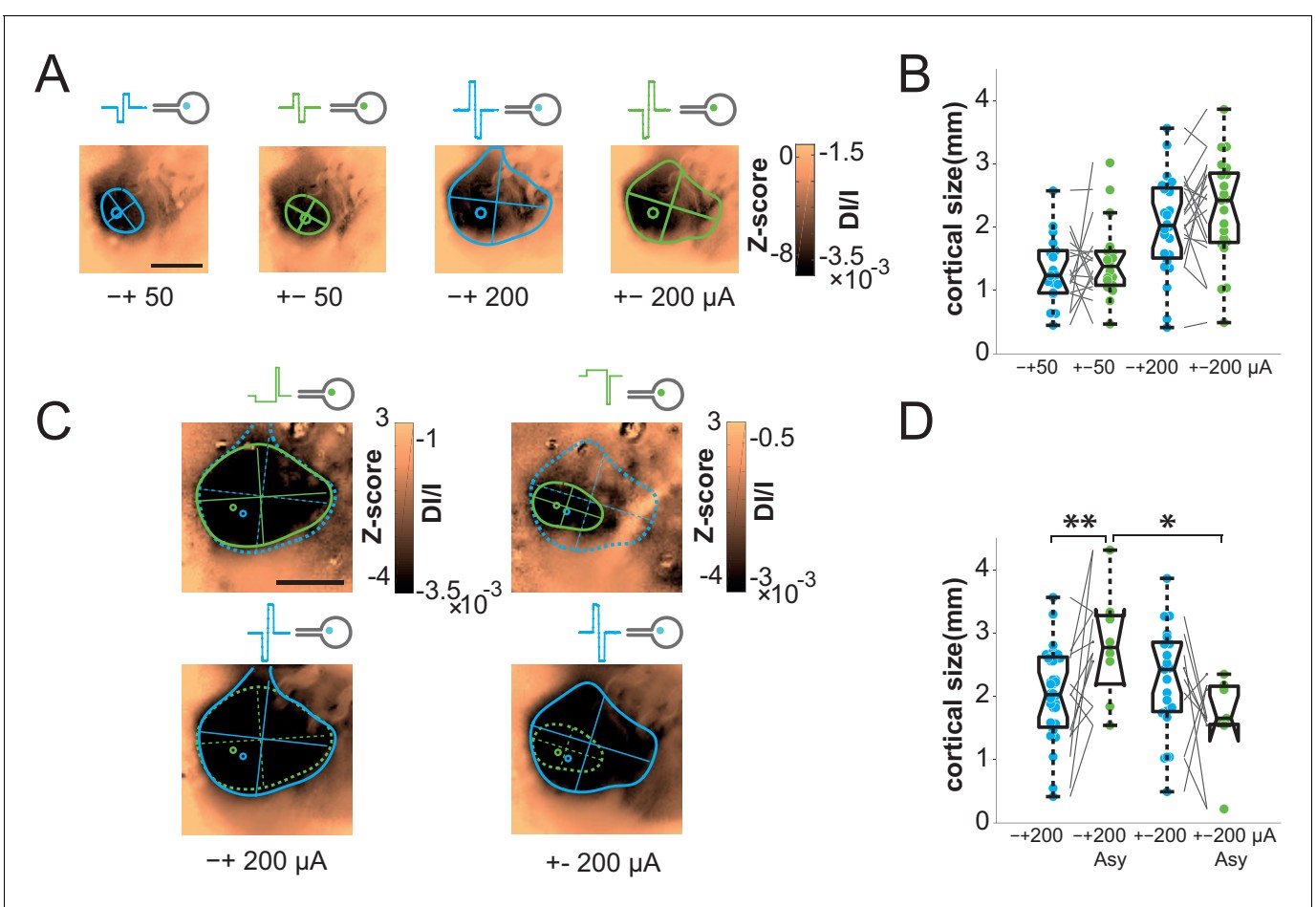

**Figure 6.** Focalization. (A) SE activations generated by square pulses of different polarity (anodic first: blue; cathodic first: green) at 2 intensity levels (50 and 200 μA). Scale bar: 2 mm. (B) Effect of polarity and intensity on cortical extent for SE (anodic first: blue; cathodic first: green, N = 10). One-sided Wilcoxon rank sum test for paired data, p=0.3802 (n = 18, N = 10) and 0.0615 (n = 11, N = 10) for 50 and 200 μA respectively. (C) Individual example of asymmetrical (green) and symmetrical (blue) square pulses, same animal as in (A). (D) Effect of square pulse asymmetry (green) for two polarities on cortical extent for SE (N = 10). One-sided Wilcoxon rank sum test for paired data: **p[CathoSym vs. CathoAsy]=0.0052 (n = 13, N = 10); *p[AnoAsy vs. CathoAsy]=0.0469 (n = 6, N = 10).

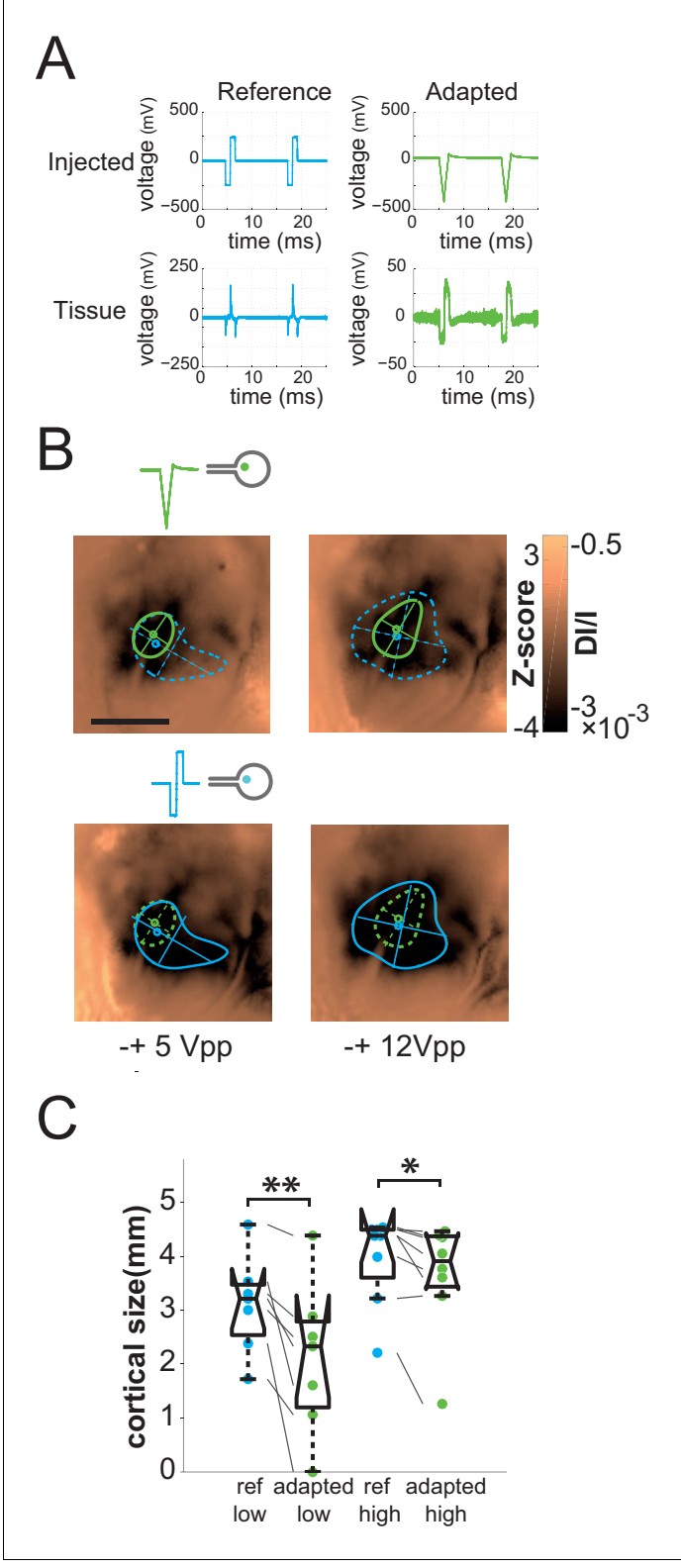

**Figure 7.** Impedance spectroscopy adaptation. (**A**) Example of electrode-tissue interface filtering on pulse shape (at 500 mVpp) with (green) and without (blue) IS based adaptation. Top: injected pulse; bottom: pulse shape reaching the tissue. (**B**) Individual example of IS adaptation (green, top maps) and reference (blue, bottom maps) for 2 voltage levels; scale bar: 2 mm. Note that for this protocol only, we switched from current to voltage injection. (**C**) Effect of IS adapted pulses on cortical extent for SE at 2 intensity levels (5 and 12 Vpp in voltage

*Figure 7 continued on next page*

*Figure 7 continued*

injection mode). IS adapted pulses: green; reference: blue, N = 4. One-sided Wilcoxon rank sum test for paired data: *p=0.0117 (n = 8, N = 4), **p=0.0078 (n = 7, N = 4).

The following figure supplement is available for figure 7:

**Figure supplement 1.** Principle of IS adaptation.

---

based (IS) adaptation (*Dupont et al., 2013*) (*Figure 7*). In the example shown and over the population (*Figure 6A–B*), our results showed that the polarity of symmetric biphasic pulses did not significantly influence the extent of the cortical activation for the two intensity levels tested. However, the polarity was found to have a significant effect in asymmetric pulses delivered at 200 μA (*Figure 6C–D*). The combination of asymmetric with anodic-first stimulation (*Figure 6C* top right) generated smaller activation of $1.66 \pm 0.71$ mm corresponding to $17.12 \pm 8.11^{oeq}$ (N = 7), smaller than the three other combinations. However, this reduction was not systematic (70% of the cases compared to the symmetrical pulse) and small ($20.43\% \pm 54.32$, the corresponding [20-50-80] percentiles being [34.78–32.93–63.89]%). Please note that this negative result is the outcome of thorough manipulation of key parameters of the electrical stimulation and highlight the high variability of the changes induced. Our best result at this stage is obtained by combining the polarity and the asymmetry parameters.

One general issue encountered when injecting current or voltage in a tissue, and often ignored, is that the desired pattern to be injected can be strongly distorted by the non-ohmic properties of the electrode-tissue interface (*Geddes, 1997*; *Pham et al., 2013*). In a previous study (*Pham et al., 2013*), we indeed showed that the impedance phase and magnitude of the electrode-retina interface was found to be highly capacitive, yielding strong distortion of the applied stimulus (*Figure 7A*, left bottom) and lateral diffusion of the injected pattern. Our rationale was therefore to use the characterization of the physical properties of the electrode-tissue interface to calibrate through reverse engineering methods what needs to be injected to obtain the desired pattern in the tissue. To do so, we characterized the impedance spectrogram of selected SE of each implantation (*Pham et al., 2013*). Using an equivalent electronic circuit accounting for the observed non-ohmic behavior, we simulated (*Figure 7—figure supplement 1B*, see Materials and methods for details on the procedure) the adapted pattern of voltage injection (*Figure 7A*, top right) that will generate the desired pattern shape in the implanted tissue (*Dupont et al., 2013*) (*Figure 7A*, right bottom). Note that, for this protocol only, we switched from current to voltage injection and that the low voltage level used here was of the same order of magnitude as the high current level used in the previous protocols. For 100% and 88% of the low and high voltage level cases respectively (5 $V_{pp}$ and 12 $V_{pp}$, $V_{pp}$: Volts peak-to-peak), the adapted pattern injected through SE decreased the extent of the activation (*Figure 7B* top maps & *7C*). This significant decrease (see *Figure 7C* legend) varied on average from $36 \pm 32\%$ to $20.7 \pm 29.2\%$ at low and high voltage, respectively (the corresponding [20-50-80] percentiles were [11.78–27.50–59.05] and [1.89–6.56–39.67]%). Importantly, at high voltage it was not accompanied by a significant change in response amplitude (Wilcoxon rank sum test: p=0.546, n = 8, N = 4), suggesting that this reduction of the extent was not simply explained by a change in activation strength level, allowing for an independent control of the extent (size) and the amplitude (intensity) of the evoked activity.

## Discussion

We have here described a detailed characterization of the functional impact of sub-retinal prosthesis at the V1 mesoscopic scale level of rats and proposed an original solution to control its extent. In detail, we have generated a precise mapping of the rodent cortical cartographical organization in response to position, size and intensity of visual stimuli. These cortical population maps were then used to establish a functional interpretation of the activations induced by electrical stimulation. The recording of robust and reliable electrically and visually evoked responses indicates that both types of information are efficiently transmitted to the primary visual cortex and prove the functional integrity of the retinal tissue after implantation, which was also anatomically confirmed by retinal imaging

(OCT). Here we demonstrate that prosthetic stimulation generates a functional activation that occurs at the expected retinotopic location and amplitude but whose extent and aspect ratio are significantly larger than expected. To control the extent of cortical activation we tested various patterns of electrical stimulation and showed that supervised design of the electrical pulses – taking into account the physical filtering properties of the electrode-retina interface – allowed focalizing the activation.

The working hypothesis of this study is that population activation of the first cortical integration stage is correlated to underlying evoked percepts, as suggested by many studies (*Chen et al., 2006*; *Ni, 2010*; *Palmer et al., 2007*; *Tehovnik and Slocum, 2006*). Furthermore, the link between population activation in optical imaging and spiking discharge has long been established (*Chen et al., 2012*; *Das and Gilbert, 1995*; *Shmuel and Grinvald, 1996*; *Toth et al., 1996*). Lastly, a recent study showed that the spatial profile of cortical population activation in the awake monkey biases the shape of the evoked percept (*Michel et al., 2013*). Therefore, we believe that the current approach is valid for an initial description of the functional activation of retinal prosthesis. The next step will be to launch experiments in behaving non-human primates, to clarify, in a model closer to humans, the relationship between the properties of the population prosthetic activation and the behavioral evaluation of the animal's percepts.

## Accurate activation in position and intensity

Prosthetic activation yielded to cortical activations that were consistent with the V1 retinotopic organization. The degree of positional precision, higher in SE compared to wMEA, was of the same order of magnitude as the 20° visual stimuli. However, this measure was quite variable, as expected given the poor precision of the murine visual system (*Euler and Wässle, 1995*). Here we significantly extended previous investigations that have first used mesoscopic optical recordings to image the cortical features of the prosthetic activation but on a limited number of animals (*Walter et al., 2005*) and with no quantitative statistical analysis (*Eckhorn et al., 2006*; *Walter et al., 2005*). Our approach allowed providing a clear and quantitative description of the prosthetic activation.

We have also shown that increasing the level of current intensity increases the amplitude of the evoked cortical response with a sigmoidal profile similar to the one observed with visual luminance. Such non-linear profiles are typically well characterized by their threshold and slope (*Naka and Rushton, 1966*). The threshold differences observed in wMEA and SE stimulation configurations can trivially result from difference in active electrode surface. However, our experiments unveiled that SE offers a larger operating range (182 *vs.* 82 µA for wMEA) to cover a theoretical modulation range of about 50 cd/m$^2$. This result demonstrates that small electrode sizes will be more appropriate to make fine manipulation of stimulus intensity. Importantly, our results also revealed that the increasing stimulus intensity increases both the response size and the response amplitude. This observation is crucial to consider since it will seriously challenge independent manipulation of those important parameters in prosthetic vision.

## Distorted activation in size and shape

Our results demonstrate that the size elicited with standard parameters is approximately 2.4 to 5.8 times larger than expected for wMEA and SE, respectively. This increase can be partly explained by passive electrical diffusion in the implant-retina interface, as we previously suggest (*Pham et al., 2013*), but also by the activation of *en passant* ganglion axons. The spatial resolution of the prosthetic retinal activation measured in our experiments, about a millimeter of retinal space, is actually comparable to what has been reported in other animals (as estimated from *Eger et al., 2005*; *Schanze et al., 2003*) as well as in human studies (as estimated from *Ahuja and Behrend, 2013*; *Humayun et al., 2012*; *Stingl et al., 2013*) and extends beyond the retinal point spread function (*Stett et al., 2000*, *2007*).

Are these results accentuated by the anesthetized state of the animal? Indeed, recent experiments showed that anesthetized mice have less surround inhibition than when awake (*Vaiceliunaite et al., 2013*). However, Vizuete et al (*Vizuete et al., 2012*) have shown that anesthesia affects not simply the inhibition but more probably the balance between excitation and inhibition (see also *Chemla and Chavane (2016)*). Hence, predicting how activation size of the population activity will be affected by anesthesia is not straightforward. Importantly, our study was devised to

make a systematic comparison between conditions done under similar level of anesthesia (visual *vs.* electrical) and the polarity of the effect (the size increasing or decreasing) is not expected to reverse by changing the animal state.

For SE stimulation, large activations were also accompanied by an increase in the aspect ratio of the spatial profile of the cortical activation suggesting an asymmetric recruitment of the retinal tissue, for instance through activation of ganglion axons *en passant* (*Nowak and Bullier, 1998a*, *1998b*). However, this increase in aspect ratio was not observed with wMEA. Using a simple model, we show that such difference in aspect ratio between stimulation conditions is actually to be expected because of differences in the spatial distribution of activation of axons en passant. Our model also allowed predicting that, to account for the observed spatial profiles, axons *en passant* should contribute to the same proportion than direct activation of the retina. In accordance with this prediction, we found that high aspect ratio activations distribute radially around the cortical representation of the blind spot and are asymmetrical, with center-of-masses displaced away from the blind spot representation. In contrast, Fransen *et al.* (*Fransen et al., 2014*) discard the possibility of direct RGCs activation through near infrared laser stimulation of photovoltaic subretinal array because of the absence of responses in the superior colliculus using inner retina synaptic blockers. However, several factors, including the low efficiency of photovoltaic transduction and the level of injected currents used in our study, might explain the differential recruitment of RGCs cell bodies and axons observed here using V1 population recordings.

Since most implantations occurred at more or less the same retinal eccentricity, our model predicts that activation size and aspect-ratio should be inversely related: the smaller the activation size (equivalent to a smaller cone angle for a fixed eccentricity), the larger the expected aspect ratio. *Figure 8* indeed shows that the averaged sizes observed for standard electrical stimulations are systematically inversely correlated to their aspect ratio. Remarkably, all regular manipulations of the stimulation pattern moved the activation spatial profile along that inverse relationship. In contrast, visual activation does not display such correlation, the aspect-ratio being small and independent of activation size (*Figure 8*, gray dots). This relationship could therefore be taken as a diagnosis of the way electrical stimulation activates the retinal circuitry with lateral isotropic diffusion and anisotropic axons *en passant* activation. To mimic a functional visual activation, it is therefore needed to decrease both the extent and the aspect ratio of the prosthetic activation.

## How to improve prosthetic activation?

Charge balanced biphasic pulses, delivered in voltage or current mode injection, are of common use in human (*Ahuja and Behrend, 2013*; *Fujikado et al., 2007*; *Humayun et al., 2012*; *Klauke et al., 2011*; *Nanduri et al., 2012*; *Rizzo, 2003*; *Shepherd et al., 2013*; *Stingl et al., 2013*; *Wilke et al., 2011*; *Zrenner, 2013*; *Zrenner et al., 2010*) or animal (*Chowdhury et al., 2008*; *Cicione et al., 2012*; *Eckhorn et al., 2006*; *Eger et al., 2005*; *Elfar et al., 2009*; *Matteucci et al., 2013*; *Nadig, 1999*; *Schanze et al., 2003*; *Walter et al., 2005*; *Wong et al., 2009*) studies of prosthetic vision. In the literature, we observed a large consensus concerning the use of cathodic pulse first for epiretinal stimulation (*Ahuja and Behrend, 2013*; *Eckhorn et al., 2006*; *Eger et al., 2005*; *Elfar et al., 2009*; *Fried et al., 2006*; *Nanduri et al., 2012*; *Schanze et al., 2003*; *Walter et al., 2005*) although not for subretinal, suprachoroidal and extraocular approaches (*Chowdhury et al., 2008*; *Cicione et al., 2012*; *Eckhorn et al., 2006*; *Fujikado et al., 2007*; *Matteucci et al., 2013*; *Stingl et al., 2013*; *Wilke et al., 2011*; *Wong et al., 2009*). It has been shown in vitro (*Jensen et al., 2005*) that the use of single cathodic pulse lowers the threshold of retinal ganglion cells (RGCs), lowers the latency of inner retina mediated RGCs response and targets more specifically RGCs cell bodies when compared to anodal pulses. As far as we know, only one study systematically examined the effect the polarity of charge balanced biphasic square pulses (*Chowdhury et al., 2008*). Using an extraocular device, Chowdhury *et al.* did not find any significant effect of the pulse polarity on the threshold of V1 electrically evoked potentials. Similarly, we did not find any effect of the polarity on the size of cortical activations. Altogether, anodic asymmetric pulses were the best among classical stimulations to restrict cortical activations, although not systematically. This result contradicts the prediction of the pioneer modeling work of McIntyre & Grill (*McIntyre and Grill, 2000*), which predicts more focal activation for cathodic asymmetrical pulses using a cable model. To conclude, basic manipulation of the electrical pulses did not yield a systematic reduction of the spatial extent of the activation. Furthermore, when compared to the aspect ratio (*Figure 8*), it turns out that all these

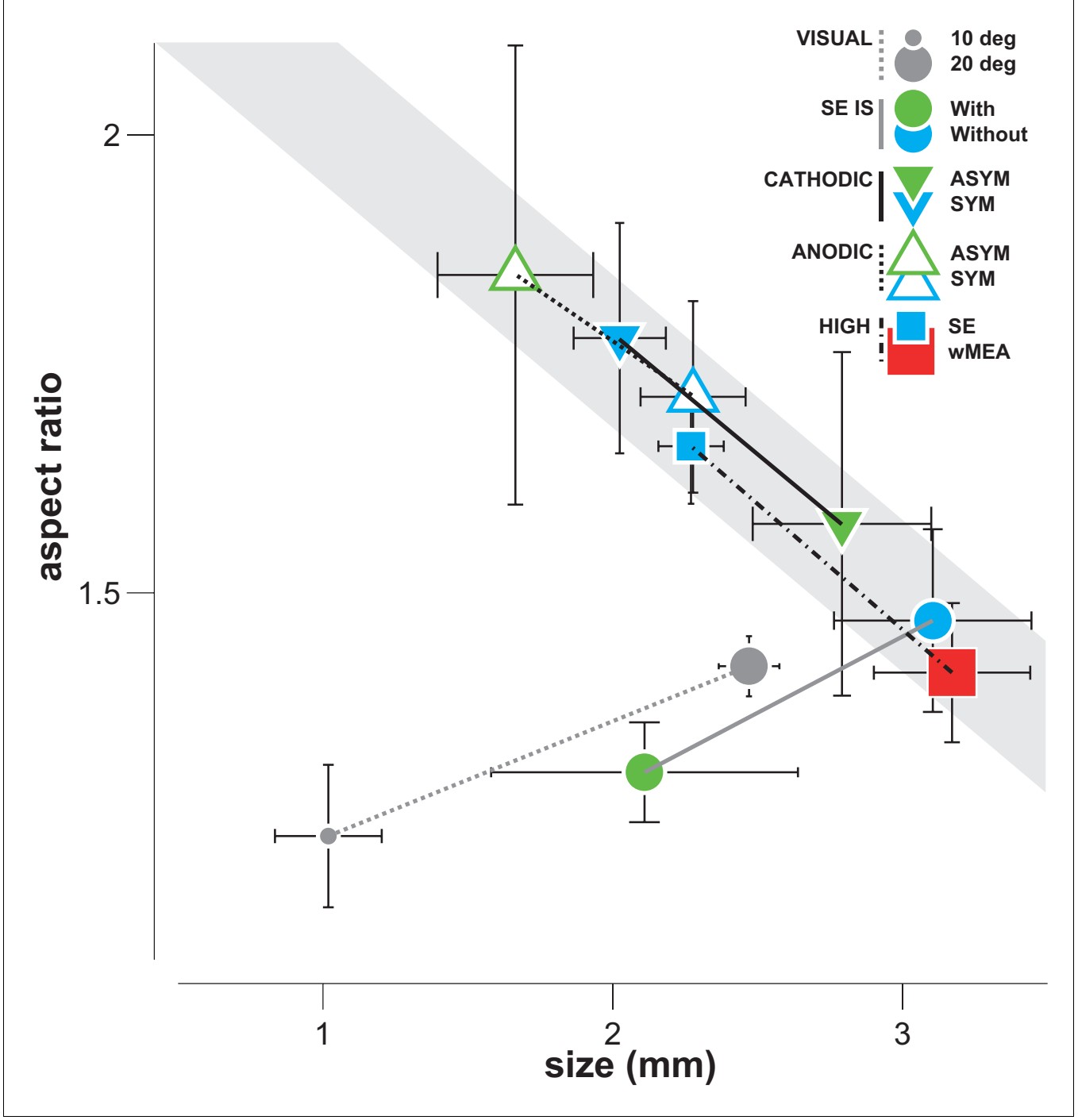

**Figure 8.** Differential effect of SE patterns. Averaged cortical size and aspect ratio (± sem: black error bars) elicited by visual stimuli, by the wMEA and by the different SE stimulation patterns delivered at high intensity levels. Gray circles: 10 and 20 deg visual stimuli (n = 7 and 75, respectively); cyan and red squares: high intensity symmetrical stimulation patterns for SE (n = 57) and wMEA (n = 10) respectively (both polarities); green and cyan circles: respectively IS adapted and non-adapted pulses delivered at 5 Vpp (n = 7, same order of magnitude as the other pulses delivered at high current intensity); green and cyan up triangles: respectively asymmetrical (n = 7) and symmetrical (n = 21) anodic first pulses delivered at ± 200 μA; green and cyan down triangles: respectively asymmetrical (n = 8) and symmetrical (n = 24) cathodic first pulses delivered at ± 200 μA. Lines link comparable conditions. The different SE patterns evoke activations that exhibit strong correlation between size and aspect ratio (gray shaded area), except for IS adapted stimuli which converge towards visual responses.

stimulation configurations lead to the same inverse relationship suggesting that all activated the retinal network through diffusion and axons *en passant*.

Note that other strategies, such as current steering methods, could be investigated in future in vivo studies (*Jepson et al., 2014a*). These authors showed in in-vitro isolated retinal preparation, that it was possible to improve the spatial resolution of the implant up to the activation of a single RGC type with a single action potential resolution (*Jepson et al., 2013*; *Sekirnjak et al., 2008*).

To further scrutinize optimal stimulation parameters, we have also used a more 'supervised' approach by adapting the shape of the desired retinal stimulation through impedance spectroscopy to account for the filtering properties of the electrode-tissue interface (resistive and capacitive components) (*Dupont et al., 2013*; *Pham et al., 2013*). Our results show that electrical stimulation adaptation systematically and accurately decreased the activation spread while preserving their location and amplitude, when compared to their paired controls. Crucially, it was the only configuration we tested whose decrease in extent was not accompanied by an increase in aspect ratio (*Figure 8*, filled circles), similarly to functional visual stimulation. This strongly suggests that such stimulation reduces both the electrical diffusion extent (although not completely) and the axons *en passant* activation. What is the origin of the reduction of electrical diffusion? This could arise from the change of the shape of the injected current, that activate less high frequencies (*Figure 7—figure supplement 1C*) that increase the spread of the electrical diffusion and the current density magnitude (*Pham et al 2013*). By which mechanisms axons-en-passant would be less activated? Two mechanisms are possible. First, this could be a consequence of the reduction of the electrical diffusion. Since we are positioned subretinally, reducing the diffusion will favor activation of elements closer to the electrode (photoreceptors, bipolar cells…) compared to further away (ganglion cells). A second mechanism would originate from the fact that the duration of each adapted pulse (anodic or cathodic) in the tissue is much longer (1.2 ms on average) than the transients contained in the non-adapted pulses (0.2 ms on average). This can favor activation of cell bodies since axons have shorter chronaxie, generally some hundreds of microseconds, than cell bodies, generally several milliseconds (*McIntyre and Grill, 2002*; *Nowak and Bullier, 1998b*, *1998a*; *Ranck, 1975*; *Stern et al., 2015*; *Tehovnik and Slocum, 2009*, *Histed et al., 2009*). This effect could be further amplified by the fact that the adapted pulse inside the tissue was also biphasic, cathodic phase first and asymmetrical in amplitude. According to the work of Mc Intyre & Grill (*McIntyre and Grill, 2002*), these are the conditions for which we can expect the threshold for activation to increase for axons and to decrease for the soma. It is thus expected that the adapted condition will lower the contribution of fiber-en-passant, since the amplitude of the current injected in the tissue was lower in adapted than in non-adapted mode. This is therefore a very promising solution getting closer to a more natural visual activation of the retinal network. Importantly, it is noteworthy to point out that this stimulation strategy can be easily embedded in already available commercial devices using small electronic circuits. Supervised adaptation of electrical pulse based on the electrical properties of the electrode-tissue interface is therefore a promising solution for retinal, but also probably, for cochlear prosthetic stimulations and more generally for all neuro-stimulation applications.

As a perspective, one may consider to combine the advantages of the different methods that have been shown, here and in other publications, to have an effect: adaptation, asymmetry of the pulse (*McIntyre and Grill, 2001*) and the spatial arrangement of stimulating and counterelectrodes (*Jepson et al., 2014a*). The possible independence of the mechanisms by which these methods decrease the activation size may allow an additive gain in spatial resolution.

## Conclusion

Here we provide a clear quantitative functional description of retinal prosthetic activations by using a systematic comparison of artificial and natural activation of the rodent visual system. Using a similar approach, further investigations will have to probe the dynamical aspect of these prosthetic activation to better understand how to generate spatio-temporal activations in the visual pathway (*Elfar et al., 2009*; *Fried et al., 2006*) that are similar to the ones observed in response to natural stimuli. A further challenge will be to drive the stimulator to generate activity closer to the retina's natural neural code (*Nirenberg and Pandarinath, 2012*), which may involve taking into account higher order correlations (*Marre et al., 2012*; *Marre and Botella, 2014*). All these strategic steps will greatly benefit from animal models such as the one used here. To conclude, our work demonstrates that pre-clinical studies are a necessary prerequisite to validate and improve the efficiency of

retinal implants, equivalent to animal models that are the foundation of any drug development for human pharmacology. We thus expect that our study will pave the way for fast and significant improvements of the clinical benefits offered to implanted patients.

## Materials and methods

### Animal preparation

A total of 35 Brown Norway male rats (1.5–3 months old, 230–320 g) were anesthetized using an intraperitoneal injection of urethane. The experimental protocol was approved beforehand by the local Ethical Committee for Animal Research and all procedures complied with the French and European regulations on Animal Research (approval n°A12/01/13) as well as the guidelines from the Society for Neuroscience. Instead of a complete craniotomy, the bone was thinned with a drill until a clear optical access to V1 surface was obtained. Finally, transparent silicone was applied to the remaining bone and the preparation was covered with a glass slide.

### Retinal implants

We used subretinal Micro Electrode Arrays (MEAs) manufactured at the CEA-LETI (*Pham et al., 2013*) (Grenoble, France). These planar MEAs of 1 and 1.2 mm diameter comprise respectively 9 and 17 (50 µm radius, 3 µm thick) platinum (Pt) contacts and a large annular Pt counter-electrode (CE) within a polyimide flexible substrate (*Figure 1C* top row). The electrical connection between the MEA and the stimulator (BioMEA, CEA-LETI) (*Charvet et al., 2010*) is made using an Omnetics 18-position nanominiature connector.

### Sub-retinal implantation procedure

The MEAs were implanted between the pigmentary epithelium and the outer segment photoreceptor layer (*Figure 1C* bottom row). Eyedrops of oxybuprocaïn chlorohydrate were used to provide a local anesthesia and the pupil was dilated using atropine drops. The bulbar conjunctiva was removed on the top of the eyeball and a millimetric incision of the sclera was performed to access the subretinal space. A cannula was then carefully inserted and a controlled injection of balanced saline solution (BSS; from B. Braun Medical Inc.) was used to induce a retinal detachment by hydrodissection. The implants were then carefully inserted and advanced below the retina towards the desired position. Eye fundus and/or OCT imaging were then performed to check implantation quality as well as the absence of potential lesions. After each implantation a recovery period from 30 min to 1 hr was respected before starting data acquisition and the presence of visual evoked cortical activation was used to probe the functional integrity of the implanted retinal tissue.

### Eye fundus and OCT

In some animals, the retina was imaged using the scanner 3D OCT-2000 (Topcon, Tokyo, Japan), allowing a complete assessment of the area around the MEA. Each acquisition combined both OCT and fundus imaging. The OCT has an axial resolution of 5 µm. The superluminescent diode light source used is centered at 840 nm with a bandwidth of 50 nm adapted for retinal imaging. The focus was adjusted manually on the retina above the MEA. The analysis was performed using 3D OCT-2000 software (Topcon, Tokyo, Japan) and consisted in the localization of the MEA on the corresponding B-Scan cross-sections. For fundus images, the OCT was combined with a camera (Nikon R_D90, Nikon Imaging Japan Inc.) that uses a white light flash with green filter, allowing color or red-free acquisitions. An additional +20 D magnifying lens was used to enlarge the field of view of the apparatus originally designed for measurements on humans. Note that the use of this lens induced optical artifacts in fundus (*Figure 1C* top row, halos of light). Finally, the positions of the implant and of the optic disk were back projected onto the visual stimulation screen using an ophthalmoscope coupled with a laser.

### Data acquisition

We imaged 5*5 mm cortical windows using 2 different intrinsic imaging systems, the MiCAM ULTIMA imaging (SciMedia, 100*100 pixels, 33.3 Hz) and data-acquisition system as well as a Dalsa camera (Optical Imaging Inc, 340*340 pixels, 30 Hz) controlled by the VDAQ data-acquisition system

(Optical Imaging). The brain was illuminated at 605 nm by 2 optic fibers. Each trial lasted 8 s and the beginning of each acquisition was triggered by the heartbeat. Five hundred milliseconds later (frame 0) the visual or electrical stimulus was displayed during 1 s. The blank condition consisted on a gray screen of 0.5 Cd/m$^2$ of averaged luminance for visual stimulation and no current injection for the electrical stimulation. The trials were repeated 20 to 40 times per condition with an ISI of 3 s.

## Visual stimulation

Stimuli were displayed monocularly at 60 Hz on a gamma corrected LCD monitor (placed at 21.6 cm from the animal eye plane) covering 100° (W) x 80° (H) of visual angle using the Elphy software (Elphy, Unic, Paris). In each trial, a gray screen with an averaged luminance of 0.5 Cd/m$^2$ was displayed during 500 ms (frame 0); the visual stimulus was then presented during 1 s followed again by a gray screen of 0.5 Cd/m$^2$ during 6.5 s. Three different protocols were then used. For retinotopic mapping, a white square (49 Cd/m$^2$) of 20° side was displayed on a gray background (0.5 Cd/m$^2$) at 20 different positions (5 horizontal*4 vertical) in a grid like manner (*Figure 2A*). For the size protocol, the same visual stimuli were displayed at 4 different positions. For each of these positions, the squared visual stimuli were displayed in different sizes. Several ranges of size were used [2 6 10 14 18°], [6 10 14 18 22°] and [3.2 9.6 16 22.4 28.8°]. In the last protocol, 20° side stimuli were also displayed at 4 different positions but with varying luminance levels ([2 13.75 25.5 37.25 49 Cd/m$^2$]).

## Electrical stimulation

Electrical stimuli were designed and injected using a BioMEA stimulator (CEA-LETI, Grenoble, France) (*Charvet et al., 2010*) in current injection mode. We used charge balanced squared pulses (1 ms per phase) of varying current intensity and polarity displayed at 80 Hz during 1 s. Such pulses were either injected through single electrodes (SE, for MEAs with 9 and 17 electrodes) or through all active electrodes (wMEA, for MEAs with 9 electrodes only) in monopolar mode. The diameters of the MEA active surface are of 50 and 650 µm for SE and wMEA, respectively which correspond to 1 and 11° according to Hughes (*Hughes, 1979*). Four different protocols were then used. In the first protocol, we tested the effect of increasing current intensity level [± 4 8 16 37.5 75 150 300 µA] (cathodic pulse first) for SE and wMEA. For the analysis, current levels ≤ 50 µA and ≥75 µA will be considered as low and high intensities, respectively. We next investigated the effect of stimulus intensity and polarity for SE only. Two current intensity levels (± 50 and ± 200 µA) and both polarities (cathodic or anodic pulse first) were used. For SE, we also tested the effect of pulse asymmetry and polarity at ± 200 µA. Instead of having symmetrical phase of 1 ms each, the asymmetrical pulses consisted of a first long phase of 1 ms at 40 µA followed by a short phase of 200 µs at 200 µA (asymmetrical ratio = 5). In total, 4 conditions were used in this protocol, 2 symmetrical and 2 asymmetrical pulses of opposite polarities. For the last protocol in SE, we switch from current to voltage injection mode to test the effect of Impedance Spectroscopy (IS) adapted stimuli as described below.

## Impedance spectroscopy measurements and adapted stimulation

Here we present a voltage-mode stimulation platform with the capability to deliver a controlled voltage pattern to the stimulation targets by taking into account the electrode-tissue interface filtering properties (*Dupont et al., 2013*). This procedure requires first measuring the impedance of the electrodes (magnitude and phase) for a wide range of frequencies (Impedance Spectroscopy). These measures unveil the capacitive nature of the electrode-tissue interface (*Geddes, 1997*). Thus the actual current or voltage pattern injected in the tissue is strongly distorted and does not resemble the desired pattern (*Figure 7A*, left column). A solution proposed by Dupont et al. (*Dupont et al., 2013*) is therefore to estimate the equivalent electrical circuit that accounts for the observed spectrum and use it to reverse engineer what needed to be used to achieve the desired injected pattern (*Figure 7A*, right column). To achieve this, the experimental design integrates an IS recording module, an Identification Algorithm module fitting IS data to an Electronic Equivalent Circuit (EEC), a Transfer Function Computing module and an Adapted Stimuli Shaping module computing adapted stimuli emitted by the Stimuli Generator module (*Figure 7—figure supplement 1A*, patent CEA-LETI) (*Dupont et al., 2013*). In this study, the test bench is built with commercial devices. IS measurements are performed by injecting low AC voltage between 100 Hz and 200 kHz with a Bio-Logic potentiostat (SP240 type). Data are fitted to a chosen EEC (*Figure 7—figure supplement 1B*, see

legend for a detailed description of the different components of the circuit) (*McAdams and Jossinet, 2000, 1996*; *Ragheb and Geddes, 1990*) using the potentiostat associated software (EC-Lab). A transfer function H linking $V_{IMPLANT}$ to $V_{TISSUE}$ is defined. To compute the adapted signal to be generated by the stimuli generator, we use the inverse transfer function $H^{-1}$ estimation. However, this transfer function does not take into account the impedance's non-linearity toward voltage levels. To address this issue, several IS measurements are performed between 1 $mV_{pp}$ and 4 $V_{pp}$ ($V_{pp}$: Volts peak-to-peak). Each set of data is fitted onto the same EEC. The abacuses 'EEC parameters *versus* voltage' are then used to compute the adapted stimulus using voltage dedicated transfer functions per signal harmonics.

## Data analysis

Stacks of images were stored on hard-drives for off-line analysis. The analysis was carried out with MATLAB R2014a (Math-Works) using the Optimization, Statistics, and Signal Processing Toolboxes. Data were first preprocessed to allow comparison between animals and conditions.

To allow comparison across conditions, sessions, animals and stimuli (visual *vs* electric) we performed temporal and spatial normalizations of optical imaging signals. We dealt beforehand with pixel-wise trial to trial variability. The averaged time-course of the outer border (2 pixels width) of the image area (outside the region of interest) was subtracted from each pixel to remove non-functional temporal noise pattern. We then performed from each trial a first temporal normalization by subtracting the mean and dividing by the standard deviation both estimated from the frame 0. Second we averaged over trials and applied a spatial normalization. Static maps were computed by averaging 1.5 to 2.5 s after stimulus onset (initial dip). Spatial z-score normalization was based on the averaged static map of the blank condition. Pixel by pixel, we subtracted from each static map the mean value of the blank and divided the outcome by the blank standard deviation over space. Significance of cortical activation was achieved by applying a threshold of $-3.09$ Z-score (activation contour) which corresponds to p-value of 0.01 (lower activations were not considered for further analysis). These activation contours were computed on a smoothed version of the maps (convolving the raw matrix with a 15x15 pixel flat matrix). To give a comparison with most optical imaging publications, we also provided on each map a second colorbar scale expressed in DI/I which was computed using the standard procedure as follows (see *Reynaud et al., 2011*): The first step consists of dividing each image of the stack by the mean of the first frames recorded before stimulus onset. In the second step, image stacks collected during stimulated trials are subtracted to those acquired during blank trials on a frame-by-frame basis.

The cortical center-of-mass was computed on the z-scored map. In order to quantify the extent and the shape of the activation contour, we computed the equivalent ellipse (*Haralock and Shapiro, 1991*). This provided the length and orientation of the equivalent ellipse minor and major axis as well as its equivalent diameter and geometric center.

In order to compute the retinotopic polar maps we computed for each pixel the visual activation map giving its response over the 5*4 visual positions. We then computed the visual center-of-mass (equivalent to a receptive field center). The abscissas or ordinates of each center-of-mass computed for each pixel across positions were plotted separately for azimuth and elevation respectively as polar map (*Figure 2B*, the colors representing the position across the 2 cardinal dimensions and the hue representing the intensity of the response for the center-of-mass position). Based on this retinotopic mapping, the expected cortical position representing each visual position was computed by taking the intersection of the 2 cardinal polar maps. For each visual or electricalevoked response, we computed the positional error as the distance between the retinotopic expected position (given by the retinoptopic position of the visual or electrical stimulus) and the center-of-mass of the corresponding activation (*Figure 2J*).

To compute cortical magnification factor, size or amplitude response functions, linear fits and Naka Rushton functions were used. For the intensity parameter, we computed the functional operational range (10–90%) for each animal individually as follow:

$$\text{operational range} = C50 * 9^{\frac{1}{n}} - \frac{C50}{9^{\frac{1}{n}}}$$

With C50 and n being the semi-saturation parameter and the exponent of the sigmoidal fit, respectively.

To investigate the spatial profile and the nature of retinal activation elicited by electrical stimulation of the MEA, we compute at the optic disk location the 'shadow cone' angle that sustains the MEA active surface as follow:

$$\text{shadow cone angle} = \frac{2 * \arcsin\left(\frac{x}{y}\right)}{z}$$

X being the MEA radius, y the distance between the MEA and the optic disk location (determined form eye fundus pictures). For each animal and stimulation condition (SE vs. wMEA), the angle values were corrected according to the real stimulated active surface diameter (z).

## Radial organization and anisotropy of activations

To find the cortical position that optimized the radial arrangement of the elongated SE cortical activations, we made the following calculation. For each cortical positions (i, j), we calculated the median value of the differences between the angles formed by the segments that connect each cortical activation center to the cortical position in question (i, j) (e.g. expected radial arrangement, dashed segments in *Figure 4E*) and the observed angle of the elongated cortical activation (solid segments in *Figure 4E*). We then looked for the cortical position for which this difference was the smallest (i.e. the cortical position for which the observed orientation of the activations was the closest to the expected radial arrangement). This gave us the cortical position that optimized the radial arrangement for our specific configuration (black disk in *Figure 4E*). We thereafter use this position rather than the median position of the optic disk (quite close, gray disk in *Figure 4E*) because we believe it is more reliable for two reasons. First, since we positioned all cortical images within the same reference frame (aligned to the midline and to *bregma*), the cortical representation of the optic disk should be approximately located at similar location in the map. Second, the individual mapping of the optic disk would have added unnecessary noise because of its imprecision (and for some sessions we could not achieve a proper mapping). Indeed, to map the optic disk representation, we estimated its position on the screen using backward projections from an ophthalmoscope. Then we put it in correspondence to the visual stimulus of our retinotopic exploration (stimuli of 20x20°) it was falling within. The center of the cortical activation for this stimulus gave an estimation of where the optic disk should be represented cortically. This introduces a series of imprecision (estimation with the ophthalmoscope, backward projection, not centered with the visual stimulus, cortical activation) that would be detrimental to our computation.

To test whether the radial arrangement of the observed orientations of SE activations occurred by chance, we used a Monte-Carlo like simulation. The question is whether the difference between the predicted (optimal radial arrangement) and the observed angles of activation is smaller than the one expected by chance. For that purpose, we generated 1000 random distributions of orientations of cortical activation at the observed positions. For each of these random distributions, we looked for the cortical position that optimized radial arrangement with the method described above. For each specific distribution that was randomly generated, we thus obtained a value of the minimum angular deviation between the predicted and the observed orientations. We obtained likewise a distribution of the angular deviation expected by chance (*Figure 4E* inset), over which we used the area under the curve of our observation to estimate the p-value: p*=0.021 (note that the distribution was also very well captured by a Gaussian fit from which we can calculate the probability of finding the observed angular deviation, and obtained a comparable z-score of 2.09, p=0.018).

## Model of retinal activation

A functional model was designed to predict spatial features of retinal activation elicited by electrical stimulation which depicted a 2D spatial layout of the retina, the central position corresponding to the optic disk. To predict the level, the shape and size of retinal activation, we have introduced two components of diffusion of activity that extend beyond the electrode surface: an isotropic Gaussian spread (G) and an 'en passant' activation (EP) of the ganglion cells' axons in response to a stimulation of a given size and a given location in the retinal space. Isotropic 'direct' activation was modeled

as a 'flat-top' Gaussian activation around the site of stimulation (see *Figure 4—figure supplement 1*):

$$IsoA(i,j) = \begin{cases} 1, & \sqrt{(i-i_0)^2 + (j-j_0)^2} < S \\ e^{\left(-\frac{(i-i_0)^2 + (j-j_0)^2}{2*(\sigma*S)^2}\right)}, & \sqrt{(i-i_0)^2 + (j-j_0)^2} \geq S \end{cases}$$

Where *i,j* are positions in space, $i_0, j_0$ the position of the electrode, *S* the size of the electrode and *σ* the spatial isotropic diffusion extending beyond the electrode.

Anisotropic *en passant* recruitment of ganglion cell axons was modeled as a shadow cone activation that would result from this 'direct' activation. To compute it, we multiply cone angle activation (*Cone*) with a sigmoid function that is 0 for distance between the blind spot and the electrode and 1 for distance further away (*Sigma*) and with a general attenuation for distances very far from the electrode (*Attenuation*):

$$EP(i,j) = Att * Sig * AngDiff \begin{cases} Att(i,j) & = e^{\left(-\frac{\left(\sqrt{(i-i_0)^2+(j-j_0)^2}-s\right)^2}{2*N^2}\right)} \\ Sig(i,j) & = \frac{1}{1+e^{\frac{dOD-(j_0-j_{od})}{S/4}}} \\ Cone(i,j) & = e^{-\frac{ConeAng^2}{2*\sigma^2}}, \quad if\ j > j_c \end{cases}$$

$$where\ dOD(i,j) = \sqrt{(i-i_{od})^2 + (j-j_{od})^2}\ and\ ConeAng = \frac{\left|\tan^{-1}\left(\frac{i-i_{od}}{j-j_{od}}\right)\right|}{2*\sin^{-1}\left(\frac{S}{j_0-j_{od}}\right)}$$

Where *Att* is a general attenuation of the activation centered on the position of the electrode (*N* is the size of the matrix), *Sig* is a sigmoid function centered on the position of the electrode ($i_{od}$, $j_{od}$ are position of the optic disk, *dOD* is the distance to the optic disk), *Cone* is a cone-like activation modeled as an angular (calculated as a normalized angle in *ConeAng*) Gaussian activation. The detail of all these computation steps is shown in *Figure 4—figure supplement 1A*.

*We* combined these two activations (*Figure 4C* insets) with different ratios (*α*) via a weighted sum (*IsoA+α\*EP*) and extracted from this predicted profiles similar parameters (size, position, elongation) as we did from our cortical recordings. In our computation, N is 400, *σ* is 1, ($i_{od}$, $j_{od}$) is (200,100), ($i_0$, $j_0$) is (200, 150), S varies between 3 and 30, and α between 0.1 and 1.

*Figure 4—figure supplement 1B and C* shows the main steps of the computation (left column is IsoA, middle column is EP and right column is the weighted sum), for two ratios (0.5 and 1, B and C) and three electrodes size (5, 7 and 20, rows).

## Statistics

To get rid of hypotheses concerning normal distribution and equal variance of data sets, as well as to be more confident in evaluating the significance of the results, we only used two-sided non-parametric Wilcoxon rank sum tests for two-sample or paired data ($*p<0.05$, $**p<0.01$, $***p<0.001$), except where otherwise stated (one-sided). The p values and sample sizes of the tests are given in the figure legends. N stands for the number of sample and N for the number of rats.

## Acknowledgements

This work was supported by the Agence Nationale pour la Recherche (ANR08-TECS-018), the Fondation Berthe Fouassier (Fondation de France, 201120574) and the Fondation pour la Recherche Médicale (DPR20121125167). The authors would like to thank F Sauter and V Agache for providing the MEAs; R Guillemaud, JF Bêche and S Gharbi for their support in the implementation of the electrical stimulation; J-M Esprabens (TOPCON, France) for providing the OCT apparatus; J Degardin (Institut de la Vision, Paris); G Sadok (UNIC, Paris); I Balansard for expert veterinary advice and help throughout the project; A Duhoux for excellent technical help; AI Meso, B Kolomiets for help in revising the manuscript. We thank B Yvert, L Nowak and A Destexhe for insightful discussions on the electrical stimulation protocols that control the fiber-en-passant activation.

## Additional information

### Competing interests

SP: Founder and consultant for Pixium Vision. The other authors declare that no competing interests exist.

### Funding

| Funder | Grant reference number | Author |
|---|---|---|
| Agence Nationale de la Recherche | ANR08-TECS-018 | Sébastien Roux<br>Serge Picaud<br>Pascale Pham<br>Frédéric Chavane |
| Fondation pour la Recherche Médicale | DPR20121125167 | Sébastien Roux<br>Frédéric Chavane |
| Fondation de France | Fondation Berthe Fouassier, 201120574 | Sébastien Roux |

The funders had no role in study design, data collection and interpretation, or the decision to submit the work for publication.

### Author contributions

SR, Conception and design, Acquisition of data, Analysis and interpretation of data, Drafting or revising the article; FM, Acquisition of data, Drafting or revising the article; FD, Designing Impedance Spectrocopy protocols, Acquisition of data, Analysis and interpretation of data; LH, Revising early stage of the manuscript, Acquisition of data; ST, Analysis and interpretation of data, Revising early stage of the manuscript; SP, Participation in early stages of the conception of the experiment, Revising early stage of the manuscript; PP, Designing Impedance Spectrocopy protocols, Analysis and interpretation of data, Drafting or revising the article; FC, Conception and design, Analysis and interpretation of data, Drafting or revising the article

### Author ORCIDs

Sébastien Roux, http://orcid.org/0000-0003-2366-6430

### Ethics

Animal experimentation: The experimental protocol was approved beforehand by the local Ethical Committee for Animal Research and all procedures complied with the French and European regulations on Animal Research (approval number A12/01/13) as well as the guidelines from the Society for Neuroscience.

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
