## [Decision Letter]

Thank you for submitting your work entitled "Probing the functional impact of sub-retinal prosthesis" for consideration by *eLife*. Your article has been favorably evaluated by David Van Essen (Senior editor) and three reviewers, one of whom is a member of our Board of Reviewing Editors.

The reviewers have discussed the reviews with one another and the Reviewing Editor has drafted this decision to help you prepare a revised submission.

Summary:

Retinal prostheses are a potentially useful therapy for blindness, but they currently provide only fairly low acuity. Roux et al. have developed an in vivo rodent preparation in order to compare visual and prosthetic activation of downstream visual areas, specifically visual cortex. They find that the cortical projective field, i.e., the cortical area activated by stimulation of single or multiple electrodes, is larger than the projective field of visual stimuli covering a similar extent of the retina. The authors then attempted to identify potential causes of and solutions to this problem, focusing primarily on spread of the electrical pulse through tissue and en passant activation of RGC axons coursing to the optic nerve. They find that by measuring the impedance of the tissue-electrode interface and correcting for this effect, they can somewhat reduce the lateral spread of excitation in cortex. This use of adaptation, particularly in tandem with other published techniques, may aid in enhancing the acuity of retinal prostheses.

Overall, the paper represents an important potential contribution to the field of retinal prostheses, but the data need to be analyzed more carefully, and the sections on cone angle and on approach to reduce lateral spread need major clarification.

Essential revisions:

1) The presentation of the cone angle model was virtually impenetrable. If the distortion is due to the radial organization of ganglion cell fibers, then the asymmetry should always be elongated radially. Was this true?

2) One expectation of en passant activation is not only an elongated RF but also an anisotropic one. They show that AR scales with eccentricity, but not whether the increased AR is asymmetric to the more eccentric side of the RF center.

3) In Figure 4 the authors show that decreasing the cone angle decreases the aspect ratio (AR). Therefore, reducing the lateral spread of activation via impedance spectroscopy adaptation should reduce the cone angle at a given point, in turn increasing the aspect ratio (AR), yet they find both parameters are reduced. It is not clear how shaping the waveforms as the authors have done would reduce en passant activation of RGC axons. Is their electrical stimulation intended to activate RGCs or photoreceptors?

4) These measurements were performed in anesthetized animals, which are reported to have less lateral inhibition in cortex. Likewise, it is unclear how the activated cortical areas correlate with acuity in awake animals.

5) The authors found multiple effects that could reduce the lateral spread of cortical activation. Do they expect or have they tested whether these effects are additive? Likewise, they cite other studies showing that interactions between adjacent electrodes can be used to narrow areas of excitation, analogous to lateral inhibition. It would be worth discussing whether these methods are compatible and how it might aid in further improving acuity with these prostheses.

6) The imaging data examples are patchy and not of the best quality, and in some cases the fitted contours do not obviously relate to the observed pattern. Moreover, the overlaid contours in many instances make it difficult to resolve the activated areas. Figure 7 illustrates both concerns, and as it pertains to the central advance the authors claim, it is not especially convincing. Nor is it clear which conditions are being shown in these images, as contours corresponding to four different conditions are overlaid on just two images. It would also be helpful to see not only Z scores but also signal magnitudes (deltaR/R) when assessing these data.

[Editors' note: further revisions were requested prior to acceptance, as described below.]

Thank you for submitting your article "Probing the functional impact of sub-retinal prosthesis" for consideration by *eLife*. Your article has been reviewed by two peer reviewers, and the evaluation has been overseen by a Reviewing Editor and David Van Essen as the Senior Editor. The following individuals involved in review of your submission have agreed to reveal their identity: Christian Casanova (Reviewer #3).

The reviewers have discussed the reviews with one another and the Reviewing Editor has drafted this decision to help you prepare a revised submission.

The reviewers think that overall the authors have addressed most of our concerns through clarifications and additional analysis, and are mostly satisfied. However, there are a few small remaining concerns:

1) Cortical magnification factor also creates anisotropy in visual receptive fields, and a worthwhile comparison would be to those of the visual projective fields in these areas, which would be the ultimate desired target of any correction.

2) It is still unclear how the aspect ratio would be lower than 1 – if there were no en passant activation, a perfectly circular patch of retina would be excited. Is this a result of cortical magnification factor? The treatment of cone angle, e.g. in Figure 4, remains unclear.

3) We appreciate the utility of Z-scores, but for the sake of others in the field who more commonly use deltaR/R and to facilitate others' comparison with their own studies, it would be useful to provide these in addition, to offer a sense of the response magnitudes observed.

---

## [Author Response]

Essential revisions:

*1) The presentation of the cone angle model was virtually impenetrable.*

Indeed, we did not describe the model in previous version of the manuscript; we now add a thorough description of the model both in the Results section (subsection “Shape”) and the Methods section (subsection “Model of retinal activation”). We also improved the supplementary figure (Figure 4—figure supplement 1) that describes the model, in link with the Methods section.

*If the distortion is due to the radial organization of ganglion cell fibers, then the asymmetry should always be elongated radially. Was this true?*

We thank the reviewers for this excellent remark. We went back on the data and analyzed the radial organization of the activations. For that purpose we kept only the cortical activations that displayed a high aspect ratio (to select those for which the elongation due to activation of axons en passant was clear). This analysis actually verified the prediction of the reviewers: we do have a significant radial organization of the activations towards the representation of the blind spot (see Figure 4). See our explanations in the manuscript (subsection “Shape”, second paragraph).

*2) One expectation of en passant activation is not only an elongated RF but also an anisotropic one. They show that AR scales with eccentricity, but not whether the increased AR is asymmetric to the more eccentric side of the RF center.*

This is a second very interesting prediction we did not tested. We are really grateful to the reviewers for these two excellent remarks that allowed us to dissect in more detail the activation profile and greatly improved the manuscript. To test for that second prediction, we compared the position of the center-of-mass to the geometrical center; the latter is only influenced by the geometrical elongation of the activation at threshold level, the first is influenced by the spatial profile of the activity. The prediction is that the center-of-mass will be attracted towards more eccentric regions of the activation (this is confirmed by our model, see Figure 9).

Author response image 1.Predictions of the distance between the center-of-mass (COM) and the geometrical center of electrical activations(normalized to implant size) as a function of the contribution of axons en passant and the distance to the optic disk (from the same model presented in Figure 4).Values >0 correspond to distances away from the BS, when compared to the geometrical center (<0 is towards the BS).**DOI:**
http://dx.doi.org/10.7554/eLife.12687.015

This is indeed what we observed systematically (see new Figure 4): all centers of masses are displaced towards more eccentric retinotopic locations. This is further confirmed when averaging (after rotation and re-centering) the shape of the cortical activations for all conditions with high-aspect ratio for which we observe that the spatial profile is asymmetric, with strong activation away from the blind spot (to the left), and with a cone shape (see inset in Figure 4). In comparison, asymmetry and cone shape are not observed for all the other activation with low aspect- ratio. Again we thank the reviewer for this very insightful comment (subsection “Shape”, second paragraph).

*3) In Figure 4 the authors show that decreasing the cone angle decreases the aspect ratio (AR). Therefore, reducing the lateral spread of activation via impedance spectroscopy adaptation should reduce the cone angle at a given point, in turn increasing the aspect ratio (AR), yet they find both parameters are reduced. It is not clear how shaping the waveforms as the authors have done would reduce en passant activation of RGC axons. Is their electrical stimulation intended to activate RGCs or photoreceptors?*

The reviewers point to an important point that was unclear in previous version of the manuscript. As we stated in the Discussion of previous version, we believe that Impedance Spectroscopy adaptation is acting “both [on] the electrical diffusion extent (although not completely) and the axons en passant activation”. Hence, this can explain how it decreases both the spread of activation (e.g. cone angle) and the AR by acting on these two mechanisms. We now further describe in the text what could be the reason for a decrease in the spread of activation. The evidence is coming from spectral analysis of the pulses: Adapted pulses have energy exclusively concentrated below 800Hz, whereas non-adapted pulses have energy ringing up to 4KHz (Figure 7—figure supplement 1). We have shown in Pham et al. (Pham et al., 2013), for good implantations) that high frequencies increase both the spread of the electrical diffusion and the current density magnitude (Figure 9A-B-C in Pham et al. 2013).

Next, we think that there are (at least) two mechanisms to account for the decrease of activation of RGC axons: First, since we are positioned subretinally, reducing the diffusion, for a given eccentricity and electrode size, will favor activation of elements closer to the electrode (photoreceptors, bipolar cells…) compared to further away (ganglion cells).

A second mechanism is that the stimulus shape itself can participate to lower the ratio of axons-en-passant activation. It is now well documented that axons have shorter *chronaxie*, generally some hundreds of microsecond, than cell bodies, generally several milliseconds (Histed et al., 2009; McIntyre and Grill, 2002; Nowak and Bullier, 1998a; 1998b; Ranck, 1975; Stern et al., 2015; Tehovnik and Slocum, 2009). If we look at the shape of the current injected in the tissue using regular non-adapted pulses (Figure 7 left row), we see that the duration of each cathodic or anodic pulse lasts about 0.23ms (Figure 7—figure supplement 1, width at half-height), hence favoring activation of axons. Our result (e.g. a ratio of en passantrecruitment of about the same contribution as direct activation) is thus not surprising. In contrast, in the adaptation condition, the duration of each (cathodic or anodic) pulse inside the tissue was much longer (Figure 7—figure supplement 1, 1.2ms), biphasic, cathodic phase first and asymmetrical in amplitude. According to the work of Mc Intyre (McIntyre and Grill, 2002), these are the conditions for which an increase in the threshold for activation of axons and a decrease for the soma are expected (see citation below*). Furthermore, since the amplitude of the current injected in the tissue was low compare to the reference condition, it is thus expected that the adapted condition will comparatively lower the contribution of fiber-en-passant.

We now include this argumentation in the Discussion of the paper (subsection “How to improve prosthetic activation?”, third paragraph) that helped clarifying this important issue. Please note that we coordinated this answer with renowned experts in the field (Lionel Nowak, Alain Destexhe and Blaise Yvert).

(*) “When an asymmetrical charge-balanced biphasic *cathodic phase first*stimulus waveform was used, there was a decrease in the threshold for activation of the local cell and an increase in the threshold to activate either passing fiber.”

*4) These measurements were performed in anesthetized animals, which are reported to have less lateral inhibition in cortex.*

Since there are not so many publications on this important topic, we believe that the reviewers point to the paper of Vaiceliunaite et al. (Vaiceliunaite et al., 2013) on mouse with isoflurane/urethane anesthesia. These authors describe in V1 neurons less surround inhibition, in other words, neurons have a maximal response for smaller stimuli when awaked than when anesthetized. This could imply that cortical activation will be larger in anesthetized condition than in the awake. However, in their paper, we can see that the overall response amplitude for large stimulus in the awake condition is still significant (see their Figure 3G & J, Vaiceliunaite et al., 2013). So, if we try to deduce from these single neuron activities the extent of cortical neuronal population significantly activated, it is not clear that the size will change, however, probably, the shape will (more peaky when awaked). This is reinforced by the evidence, in the rat, that both excitation and inhibition are affected by anesthesia (Vizuete et al., 2012) suggesting that surround suppression decrease could be accompanied by a drop in excitatory level as well. In a recent model, we investigated how the population network response is affected by anesthesia (Chemla and Chavane, 2016), and we confronted our prediction to experimental measure. Our conclusion was that the whole excitatory/inhibitory balance was indeed disrupted. When taken globally, this suggests that it is straightforward to make a simple prediction of how anesthesia will change the activation extent. A direct measure is warranted. We actually did a series of experiments on the monkey using optical imaging (Chemla and Chavane, 2016), comparing the response dynamic in awake vs. anesthetized conditions (where we confirm the general decrease in response amplitude documented in the mouse and rat in the two previous articles). From one of these experiments, not used in the article, we presented a local stimulus (1° in diameter, optimized in spatial and temporal frequency) in the same monkey while he was awake, and when he was anesthetized using midazolam (potentiates GABAA, see Figure 10). In this experiment, we show that the population response is actually smaller in the anesthetized condition (reduction of 14% of the equivalent diameter of the ellipse formed by the contour taken at 75% of the response maximum, hence lowers absolute threshold for the anesthetized condition). As a conclusion, it is not trivial to predict how the cortical spatial profile of activation will be affected by anesthesia. However, this was an important point to clarify.

Importantly, the main rationale of our study was to make a systematic comparison between conditions done under similar level of anesthesia (visual vs.electrical). It would be very surprising to observe (and we actually cannot see any mechanism that would subtend it) an inversion of the effects when shifting from anesthesia to awake state (i.e. reduction of response size from condition A to condition B would become increased by changing the animal state). We added a paragraph in the Discussion to comment this point (subsection “Distorted activation in size and shape:”, second paragraph).

Author response image 2.V1 cortical activation measured with voltage sensitive dye imaging in non-human primate, elicited by local drifting gratings in awake (left) and anesthetized (right, midazolam) condition.Red and white contours represent 75 and 50% of maximal activation; scale bar: 1 mm. Drowsy state of the animal (right) was induced by intranasal administration of 0.2 ml midazolam (7-chloro-imidazo, 0.1 mg/kg). Midazolam is a short-acting hypnotic-sedative drug with anxiolytic and amnestic properties. From the benzodiazepines class of tranquilizer drugs, the pharmacologic effects of midazolam are mediated through a majority of GABAA receptors (see Sigel 2002 for a review).**DOI:**
http://dx.doi.org/10.7554/eLife.12687.016

*Likewise, it is unclear how the activated cortical areas correlate with acuity in awake animals.*

In the previous version of the manuscript, we already discussed the relation between population activity and evoked percepts: “The working hypothesis of this study is that population activation of the first cortical integration stage is related and proportional to underlying evoked percepts, as suggested by many studies (Chen et al., 2006; Ni, 2010; Palmer et al., 2007; Tehovnik and Slocum, 2006). […] The next step will be to launch experiments in behaving non-human primates, to clarify in a model closer to humans, the relationship between the properties of the population prosthetic activation and the report of the animal’s percepts.”

However, the reviewer points to a supplementary and important issue: is the size of the activation directly linked to the size of the percept? This is what is expected in first simple assumption: a large activation in the optical imaging map will generate a large retinotopic spiking region that can be decoded as high probability of a large stimulus being present in the outside world. In a recent study, the group of E. Seidemann actually showed, using optical imaging, that the shape of the cortical activation in V1 bias accordingly the shape of the evoked percept (Michel et al., 2013), suggesting that there is a link between the spatial profile of the activation and the percept. Further evidence comes from the publications from Tehovnik and Shiller group using cortical stimulation. Schiller et al. (Schiller et al., 2011) indeed show that by increasing stimulation intensity of the cortex, there is a tendency for monkeys to perceive the stimulus size larger. This should be taken in consideration with the result of a previous publication, where Tolias et al. (Tolias et al., 2005) have shown that increasing the stimulus intensity increases the size of the cortical activation. These are two evidences that the spatial profile of cortical activation in V1 is most probably linked to the size of the percept. We now add this in the Discussion (second paragraph), arguing in favor of pursuing these studies in awake animals, using behavioral tasks, in order to further demonstrate this important link.

*5) The authors found multiple effects that could reduce the lateral spread of cortical activation. Do they expect or have they tested whether these effects are additive?*

We haven’t tested whether the effects are additive unfortunately. The two parameters that seem to affect the activation extent from our study are adaptation and asymmetry.

Note that the adaptation itself affected the asymmetry of the pulses in the tissue in the relative amplitude between the two phases but not in duration. However, duration of the asymmetry could also contribute in increasing the threshold of activation in axons (see (McIntyre and Grill, 2000), and our discussion on “focalization”). This is the reason why we tested it; unfortunately the effect was not systematic. Interestingly, diffusion in itself should not be much affected by asymmetry, because the main factor is the distance of the electrode to the retinal tissue (Pham et al., 2013).

Hence asymmetry in pulse duration could be a parameter that acts on the threshold of activation of fiber-en-passant through a complementary and sufficiently independent mechanism to generate an effect that will be additive to the one generated by the adaptation protocol. We now add comments on this in the last paragraph of the subsection “How to improve prosthetic activation?”

*Likewise, they cite other studies showing that interactions between adjacent electrodes can be used to narrow areas of excitation, analogous to lateral inhibition. It would be worth discussing whether these methods are compatible and how it might aid in further improving acuity with these prostheses.*

We agree with the reviewers. The spatial arrangement of stimulation electrodes (passive or active return electrode close to stimulating electrode) is a key factor that would be worth testing. The current will be more localized between the stimulating and the return electrode. This is another independent parameter that could be combined with adaptation of pulse.

So we would end up with three potential independent factors to be combined: adaptation, asymmetry, and spatial arrangement of the electrodes. To test this, however, would require a high combination of tests. We now add comments on this in (subsection “How to improve prosthetic activation?”, last paragraph).

Please note at this stage that we did not do justice to the difficulty of the adaptation experiment. For these latter, the experimental bench was composed of an assemblage of commercial devices (signal generator (Agilent 33250A) + impedance spectroscopy potentiostat (Bio-Logic SP200) and the estimation of each adapted pulse for each electrode and each voltage level was performed externally using Matlab (see Dupont, 2014). This generated important delays in the experiments that prevent testing a large set of parameters. Further experimentations to test the combination of adaptation, asymmetry and spatial arrangement, to be performed on same animals, will necessitate setting up an experimental bench where adaptation is performed in real-time. This implies a series of further experiments that we are trying to set but that are not trivial. All those propositions are underlining the complexity of the approach, because of the large number of parameters to explore and the amount of experiments to be performed for a proper statistical study. This was probably not fully clear in our original manuscript and we clarify it now in the first paragraph of the subsection “Focalization”).

However, we believe that our study is yet an important step for the field of retinal prosthesis because (i) it is the first that tried to explore systematically a sub-part of the parameter space, (ii) critically takes into account for the bias generated by the electrode- tissue interface properties by direct measure of the injected current and (iii) propose an original inverse engineering approach for the control of the desired stimulation.

*6) The imaging data examples are patchy and not of the best quality, and in some cases the fitted contours do not obviously relate to the observed pattern. Moreover, the overlaid contours in many instances make it difficult to resolve the activated areas. Figure 7 illustrates both concerns, and as it pertains to the central advance the authors claim, it is not especially convincing. Nor is it clear which conditions are being shown in these images, as contours corresponding to four different conditions are overlaid on just two images. It would also be helpful to see not only Z scores but also signal magnitudes (deltaR/R) when assessing these data.*

The reviewer is right. We probably oversimplified the figures by putting too many information in a single map. Second, the gray scale color-code we chose was probably not the best color-code to see variations of the activation in the map. Lastly, we forgot to mention that the contour was calculated on a smoothed version of the maps (now added in the Methods, subsection “Data analysis”, second paragraph). We now provide a single map for each condition and overlap contours to ease comparison, a secondary DI/I scale, and use another color- code that we believe provide more perceptible gradation of activity level. Regarding DI/I vs. z-score, we agree that DI/I is a scale of reference so important to keep for comparison. However, please note that the z-score measure was devised to ensure proper comparison of maps across conditions (see Jancke et al., 2004). The importance of this calculation is grounded because of experimental noise and variation across experiments, the baseline and the amplitude of the DI/I are highly variable compare to the z-score values (see Figure 2 and even more in Figure 3). Comparison between these maps would be thus impossible. The z-score maps provided a common statistical metric that was crucial to ensure comparative analysis.

[Editors' note: further revisions were requested prior to acceptance, as described below.]

*The reviewers think that overall the authors have addressed most of our concerns through clarifications and additional analysis, and are mostly satisfied. However, there are a few small remaining concerns:*

*1) Cortical magnification factor also creates anisotropy in visual receptive fields, and a worthwhile comparison would be to those of the visual projective fields in these areas, which would be the ultimate desired target of any correction.*

The reviewers are right: magnification factor can induce anisotropy in population receptive fields. This is actually exactly the reason why, on Figure 4, we show the aspect ratio (AR) of the electrically-induced cortical activations normalized by their visual-counterparts. This allowed correcting for anisotropy due to cortical magnification factor, but also variability of the activations’ shape (due to vasculature artifacts for instance).

To further answer the reviewers’ concerns we looked at two aspects of this transformation:

A) We checked whether there is any systematic anisotropy of the visual activations due to magnification factor that could be taken into account in our analysis. The prediction drawn by retino-cortical transformations is that, at eccentric locations, cortical activations should get elongated tangential to the radial axis (see the simulated “cortical space” in Figure 11). We analyzed whether such effect existed systematically in our data. The results (see Figure 11) show that the elongations of visual responses are too variable to be used as a predictor. Indeed only activations with high AR (long segment) do show some tangential organization as predicted. However, if we restrict our analysis to high AR, this would drastically decrease the data points. Therefore, we believe that the ratio of the electrical AR to visual AR is the simplest yet reliable measure to do.

Author response image 3.(**A**) Simulated deformation of cortical activations (right) to circular retinal regions located at various eccentricities and angles (left), the center of the gray pattern corresponding to the optic disk location. For this simulation, we used our estimation of the retino-cortical magnification factor. (**B**) Orientations of visual activations on the cortical surface. Length of the segments is proportional to the aspect ratio of the activation. Gray disk represents the median blind spot representation, scale bar: 0.5mm.**DOI:**
http://dx.doi.org/10.7554/eLife.12687.017

To avoid confusing the readers, we propose not to mention this in the article. We now simply argue in the Results section: “Please note that, to account for any potential deformation of the evoked activity due to retino-cortical magnification factor (see Figure 4—figure supplement 1) or physiological noise, we normalized all electrically-induced AR to their corresponding visual AR”.

B) Second, we checked directly on our model whether the retino-cortical transformations could further affect our initial prediction. This is shown in the figure below, that we propose to add as a supplementary figure. The retino-cortical transform does not change the result qualitatively: In panel A we show the mapping transformation we used for our model. In panel B we give 4 examples of the transform for different activation sizes and ratios between direct activation and axons-en-passant. In panel C, we show the same calculation as we did in Figure 4, but for the cortical prediction. Here, to achieve a similar analysis as in Figure 4, we show the normalized aspect ratio (normalized to what was obtained when we removed the contribution of axon-en-passant) as a function of implant size. Qualitatively, the result is not changing when applying this transformation (if we don’t normalize the results are also not affected qualitatively).

The result is nice and reassuring but we prefer to keep it as a supplementary figure, since this part of the model is adding unnecessary complications and parameters for an effect which is already present, and originate at the retinal level. We now add in the “Please note that these predictions are left unchanged if we apply a retino-cortical transformation to the simulated activations (see Figure 4—figure supplement 1).”

*2) It is still unclear how the aspect ratio would be lower than 1 – if there were no en passant activation, a perfectly circular patch of retina would be excited. Is this a result of cortical magnification factor? The treatment of cone angle, e.g. in Figure 4, remains unclear.*

Our apologies if it remained unclear. First, we realized we actually did not explain the convention we chose to compute the AR in the model. Since we wanted to extract elongation along the radial axis, we used as numerator the length of the activation along the radial axis and as denominator, the length of the activation along a tangential axis. We wrote in the revised manuscript: “Since our interest is to predict the effect of axon-en-passant activation on radial elongation, we used the following convention for the AR: the numerator is the length of activation along the radial axis and the denominator is the length of activation along a tangential axis (perpendicular to the radial axis).”

Second, in our model, the aspect ratio is getting lower than 1 when both the cone angle and axons en passant recruitment are large (Figure 4). When the cone angle become large (for instance when the whole MEA is stimulated very close to the blind spot), the radial diffusion of activity tangential to the radial axis becomes more important than the radial activation. This leads to an elongation tangential to the one observed with small cone angle, hence an AR lower than 1. In our simulation, this inversion is observed around 40°, a value for which the tangent of the cone angle (i.e. tangential activation) becomes larger than the cosine (radial activation). This effect is strengthened by axon-en passant contribution that will accentuate the diffusion radially and tangentially. Thanks to the reviewer comment, we further analyzed our data to verify whether this prediction was indeed verified experimentally. We now include a third averaged cortical activation in Figure 4 for the case of wMEA activations (*). In this averaged cortical map, we indeed see that the activation in response to wMEA is, as predicted by the model, more elongated along an axis orthogonal to the one for SE.

We added the following to the revised manuscript: “The model further predicts that the retinal activation should become more elongated along a tangential axis for very large cone angle (AR<1).”

“Thus, the predictions that arise from this observation are: for small cone angle (i) the elongation of the cortical activation should be radially organized towards the representation of the optic disk for small cone angle and (ii) the activation should be anisotropic, attracted by the radial elongation towards more eccentric positions (i.e. away from the optic disk representation); lastly (iii) for large cone angle, cortical activation should be more elongated along a tangential axis.”

“For comparison, we made similar averaging on all the other maps in response to SE (with an aspect-ratio < 1.6) and wMEA. For SE activation with low AR, the result shows an activation that is more isotropic. For MEA activation, the averaged map shows, as predicted, a slight elongation along the tangential axis.”

We also clarified the presentation of Figure 4.

(*) Please note that we also improved those maps, they are now oriented, not according to the orientation of their main elongated axis as presented in previous revision, but according to the radial axis linking activation to BS, the BS being on the right of the figure.

*3) We appreciate the utility of Z-scores, but for the sake of others in the field who more commonly use deltaR/R and to facilitate others' comparison with their own studies, it would be useful to provide these in addition, to offer a sense of the response magnitudes observed.*

To further clarify this point, and in addition to Figure 4—figure supplement 1 of our previous revision, we computed the correlation coefficient r^2^ between all pixels of Z-score vs DI/I maps over 225 maps (which correspond to all the data recorded on the animals presented in Supplementary Figure S2: N=9 rats, n=225 conditions). We found a highly significant*** correlation (see Figure 12, all 225 p_val_= 1.40e^-45^) between these two measures with a median r^2^ of 0.81 and the [20-50-80] percentiles being [0.73-- 0.81--0.88]% . Thus, the two types of maps are highly spatially correlated. As representative examples, we provide in the Figure 12 the DI/I version (with a double scale bar expressed in DI/I and in Z-score values) of the first maps shown in Figure 2 and Figure 2—figure supplement 1. We added these values into the text p:5, l:142.Hence, since there is a very good correlation, we believe that the z-scored maps shown with a colorbar for the z-score value (left of the colorbars) together with the corresponding DI/I min and max values (right of the colorbars) provide the reader with a very good handle to compare our data with results obtained in other studies.

Author response image 4.Correlation between DI/I and Z-score values.(**A**) Histogram of highly significant*** r^2^ correlation coefficients between all pixels of Z-score vs DI/I maps (median r^2^= 0.81, N=9 rats, n=225 maps, all p_val_= 1.40e-45). (**B**) Representative examples of DI/I maps (same as the one presented in Figure 2 and Figure 2—figure supplement 1).**DOI:**
http://dx.doi.org/10.7554/eLife.12687.018